



# Detecting seasonal ice dynamics in satellite images

Chad A. Greene[1], Alex S. Gardner[1], and Lauren C. Andrews[2]

[1]Jet Propulsion Laboratory, California Institute of Technology, Pasadena, CA 91109, USA
[2]NASA Goddard Space Flight Center, Greenbelt, MD, USA

**Correspondence:** Chad A. Greene (chad@chadagreene.com)

**Abstract.** Fully understanding how glaciers respond to environmental change will require new methods to help us identify the onset of ice acceleration events and observe how dynamic signals propagate within glaciers. In particular, observations of ice dynamics on seasonal timescales may offer insights into how a glacier interacts with various forcing mechanisms throughout the year. The task of generating continuous ice velocity time series that resolve seasonal variability is made difficult by the finite integration time over which ice velocities are measured from optical and repeat SAR imagery, and by a spotty satellite record that contains no optical observations throughout dark, polar winters. In this paper, we describe a method of analyzing feature-tracked velocities to characterize the magnitude and timing of seasonal ice dynamic variability. Our method is agnostic to data gaps and is able to recover climatological average winter velocities regardless of the availability of direct observations during winter. Using characteristic image acquisition times and error distributions from Antarctic image pairs in the ITS_LIVE dataset, we generate synthetic ice velocity time series, then apply our method to recover imposed magnitudes of seasonal variability within $\pm 1.4$ m yr$^{-1}$. We then validate the techniques by comparing our results to GPS data collected on Russell Glacier in Greenland. The methods presented here may be applied to better understand how ice dynamic signals propagate on seasonal timescales, and what mechanisms control the flow of the world's ice.

## 1 Introduction

Earth-observing satellites have been in orbit for over half a century, but it was only in 2011 that a sufficient quantity of data had been collected to complete the first pan-Antarctic map of ice velocity (Rignot et al., 2011). Since then, new satellites have led to follow-on mappings that identified regions of changing ice flow (Gardner et al., 2018), and today data are being collected at such a rate that velocity mosaics can be generated every year for nearly all of the world's ice (Joughin, 2017; Mouginot et al., 2017; Gardner et al., 2019). So for a field of study that was born in an age of *in situ* stake networks and dead reckoning, the data revolution of the past decade has completely upended glaciology. Where once our challenge was to squeeze as much information as possible from a few sparse field measurements, our biggest challenge now lies in processing massive, often unwieldy datasets, and finding all the meaningful signals that lie hidden in this new abundance of data.

One of the most direct and insightful ways to understand how ice moves, what controls its flow, and how it responds to changes in its environment, is to observe dynamic variability under a wide range of periodic forcings. Long-term trends and interannual variability (e.g., Moon et al., 2012; Christianson et al., 2016; Greene et al., 2017a; Dehecq et al., 2019) can provide a sense of how glaciers respond to sustained forcing, but on much shorter timescales we are able to see how accelerations

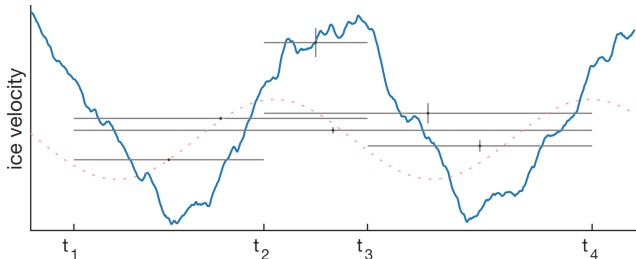

**Figure 1. Example scenario:** For a true continuous ice velocity time series like the one shown in blue, feature tracking measures velocity as the total displacement that occurs between satellite images of the flowing ice. Here, four images taken at times $t_1$ through $t_4$ provide six unique combinations of image pairs that yield six velocity measurements, which are depicted as horizontal gray lines connecting the acquisition times of each image pair. Vertical gray lines show measurement uncertainty and a black dot is placed at the center times of each image pair for visual clarity. The pink dashed sinusoid is fit to velocity measurements at the center times of each image pair to highlight the inadequacy of fitting directly to velocity data to determine the amplitude or phase of seasonal variability. This paper describes an alternate, exact approach, wherein sinusoids are fit to accumulated displacements rather than velocity time series.

initiate and what physical processes control a glacier's movement. Several targeted studies have shown that glaciers can exhibit observable dynamic responses to ocean tides, and that tidal signals can propagate well inland of the grounding line on daily to fortnightly timescales (e.g., Anandakrishnan et al., 2003; Walter et al., 2012; Rosier and Gudmundsson, 2016; Minchew et al., 2017; Robel et al., 2017), but in many glaciers around the world, a significant gap exists in our understanding of how ice dynamic changes develop between tidal and interannual timescales.

It is certainly understood that many mountain glaciers speed up and slow down throughout the year (Burgess et al., 2013; Armstrong et al., 2017; Yasuda and Furuya, 2015; Kraaijenbrink et al., 2016), and that some of Greenland's glaciers respond to seasonal cycles of subglacial hydrology or calving dynamics (Joughin et al., 2008; Howat et al., 2010; Bartholomew et al., 2010; Sole et al., 2013; Moon et al., 2015; King et al., 2018). Seasonal variability has even been reported in a few studies of Antarctic glaciers (Nakamura et al., 2010; Zhou et al., 2014; Greene et al., 2018); but to date, no global-scale mapping of seasonal dynamics of the world's ice has been completed, due in part to the logistical challenge of working with optical data in polar regions, where the surface is not touched by sunlight for months-long periods each winter.

In this paper, we describe a method that allows for the robust extraction of seasonal changes in ice flow, and may be used to map the magnitude and timing of the seasonal dynamics of all the world's ice. Our study is primarily focused on Antarctica, where seasonal variability is poorly understood, and where data limitations currently present the greatest challenges to making such measurements. We test the sensitivity of our method on several thousand synthetic ice velocity time series, then validate it by applying the method to satellite data covering Russell Glacier in Greenland, where we compare our results to GPS observations that show persistent seasonality.



## 2  Feature-tracked velocity data

The method we present applies to ice velocity datasets such as GoLIVE (Scambos et al., 2016) or ITS_LIVE (Gardner et al., 2019), which have been derived by feature tracking techniques applied to satellite image pairs. For a detailed review of the principles of feature tracking we refer readers to Scambos et al. (1992) or Fahnestock et al. (2016), but for the work presented here it is essential to know only that feature-tracked velocities are measured as the integrated surface displacement that occurs between the acquisition times of two satellite images of a given location. That is, each measurement represents an *average* velocity between image acquisition dates, and the required passage of time between images precludes direct measurement of instantaneous velocity at any given time. As a result, any high-frequency variability that occurs between images is not represented, and seasonality may appear missing or deprecated in velocity measurements obtained by feature tracking (Fig. 1). Nonetheless, by fitting a cyclic function to the time series of displacements rather than average velocities, we show that it is possible to recover the true magnitude and phase of seasonal velocity variability.

This paper focuses primarily on the Landsat image pairs that populate the ITS_LIVE dataset, in part because their record extends back as far as 1985 in some locations. The long Landsat record may help ascertain a climatological seasonal cycle of ice dynamics at any given location; however, we face the limitation that optical satellites like Landsat do not collect data throughout the dark winters in polar regions, where land ice is most prevalent (see Fig. 2). Despite the lack of direct observation in winter, we will show that the magnitude and timing of seasonal variability can be accurately retrieved from Landsat data, regardless of when in the year the maximum velocity occurs.

For any given 240×240 m pixel in Antarctica, the ITS_LIVE dataset may contain from a few dozen to more than 10,000 velocity measurements (see Fig. 3), which are taken as the ice displacement observed between two satellite images collected on different days. Each satellite image may serve as the first or second image in multiple image pairs, resulting in many overlapping measurements of ice velocity, as shown in Fig. 4. Georegistration error of each image leads to some visible disagreement between the overlapping velocity measurements, but despite the noise, a coherent pattern of interannual variability is apparent as the clusters of velocity measurements move up or down from year to year.

## 3  Method of analyzing image pair velocity data

### 3.1  Assess and remove interannual variability

The first step toward quantifying seasonal variability for any pixel is to remove any interannual variability from the time series. Interannual variability can be determined by smoothing the velocity data using any of several common methods. A polynomial fit is robust and computationally efficient, but requires choosing a polynomial order, which is subjective and can lead to overfitting or underfitting the data. Alternatively, a moving average makes no prior assumption about the shape of the velocity curve, and can adapt to any arbitrary interannual variability. After exploring several approaches, we find the best results by first detrending the time series with a low-order polynomial, then using a hybrid of a moving average and a spline fit, which we describe below.

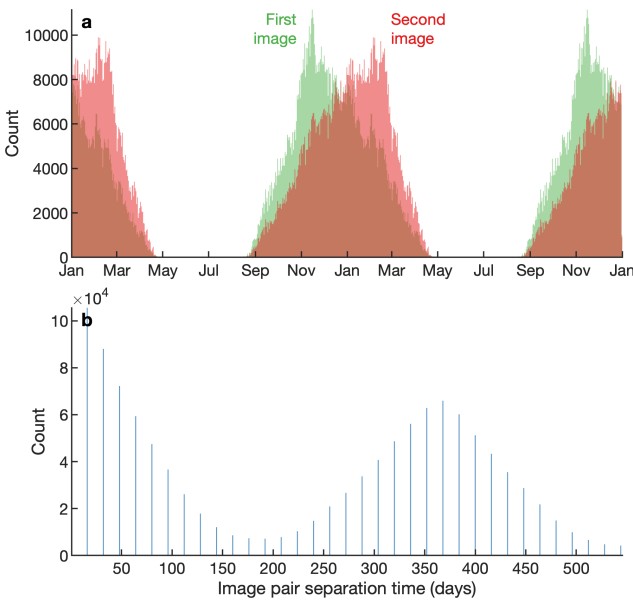

**Figure 2. Antarctic image pair acquisition times.** Over 1.8 million Landsat image pairs provide Antarctic ice velocities in the ITS_LIVE dataset. **a:** The seasonal cycle of Landsat image collection shows the effects of the solar cycle on Antarctic sampling. The cycle is repeated for visual continuity over the summer. **b:** Displacement fields in the ITS_LIVE dataset are processed for all image pairs separated by 16 to 544 days, with half of all image pairs representing 80 days of displacement or less, but a distinct secondary peak corresponds to a $\Delta t$ value of 1 year.

When assessing interannual variability, we temporarily ignore the duration over which each image pair measured ice displacement, and simply assign the average velocity to the center date $t_\mathrm{m}$ of each image pair. Using the range of times $t_\mathrm{m}$ in the time series, we require at least two years of velocity data and detrend using a polynomial whose order is chosen as one quarter
of the range of $t_\mathrm{m}$ in years, rounded up to the nearest integer. The result is that for up to four years of data, the time series is linearly detrended; from four to eight years of data, the time series is quadratically detrended, and so on. We assign the velocity weights $w_\mathrm{v}$ in the polynomial fit using the formal error estimates $\sigma_\mathrm{v}$ from the ITS_LIVE data such that $w_\mathrm{v} = \sigma_\mathrm{v}^{-2}$. An example of a fourth-order polynomial fit to the velocity data is shown in Fig. 4.

After detrending the time series with a weighted polynomial fit, we characterize any residual interannual variability with a
spline fit to the mean velocities of each year. We take the weighted mean velocity of all measurements whose $t_\mathrm{m}$ lies within 183 days of winter solstice of that year. We assign the weighted mean velocity of each year to the weighted mean date of those velocities, then interpolate with a shape-preserving piecewise cubic hermite polynomial to obtain a measure of interannual variability corresponding to each image pair's center date $t_\mathrm{m}$.

In the method described thus far, most summer image pairs whose $\Delta t$ is small contribute much less to the weighted mean
annual velocities than do long $\Delta t$ image pairs that span winter. This is because velocity error by any feature-tracking algorithm stems strictly from displacement error $\sigma_\mathrm{d}$, while timing error is essentially zero. So although summer offers more image





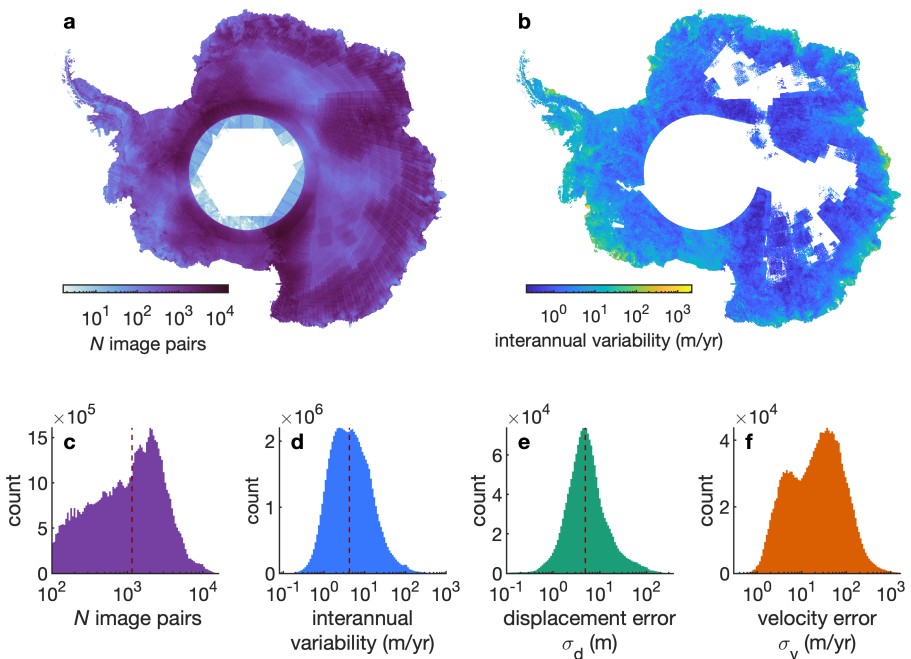

**Figure 3. ITS_LIVE velocity data statistics.** Maps of **a:** the number of image pairs that contribute to each $240 \times 240$ m pixel in the ITS_LIVE summary velocity mosaic and **b:** the standard deviation of annual velocity mosaics from 2013 to 2018. Histograms in panels **c** and **d** show only the values from **a** and **b** that correspond to grid cells with at least 100 contributing image pairs and whose mean velocity is at least $15 \mathrm{\,m\,yr^{-1}}$. Displacement errors $\sigma_\mathrm{d}$ shown in panel **e** result from satellite image georegistration error, and when divided by values of $\Delta t$ corresponding to each image pair results in the velocity errors $\sigma_\mathrm{v}$ shown in panel **f**. Because displacement errors are distributed symmetrically, the bimodal distribution of velocity errors roughly correspond to the two peaks of $\Delta t$ values over which velocity is measured (shown in Fig. 2). Vertical lines in panels **c–e** indicate median values of 1153 image pairs per grid cell, $4.2 \mathrm{\,m\,yr^{-1}}$ interannual variability, and 4.9 m displacement error.

pair measurements, their shorter $\Delta t$ values are associated with greater velocity error $\sigma_\mathrm{v}$ and therefore they are significantly downweighted in the calculation of annual mean velocities. In other words, it is possible that either due to the higher weights of winter velocities or the higher quantity of measurements available during summer, the weighted mean annual velocities could

be biased toward one season or the other. We account for this possibility by iteratively solving for interannual and seasonal variability, as we describe later in Section 3.3.

## 3.2   Assess seasonal variability

We characterize the magnitude and timing of seasonal variability as a simple sinusoid that can be applied in the $x$ and $y$ directions separately to build a two dimensional understanding of how ice moves throughout the year. Limitations of the

sinusoidal approximation are discussed later in Sect. 6, but we justify our approach as it is the simplest, most robust means of

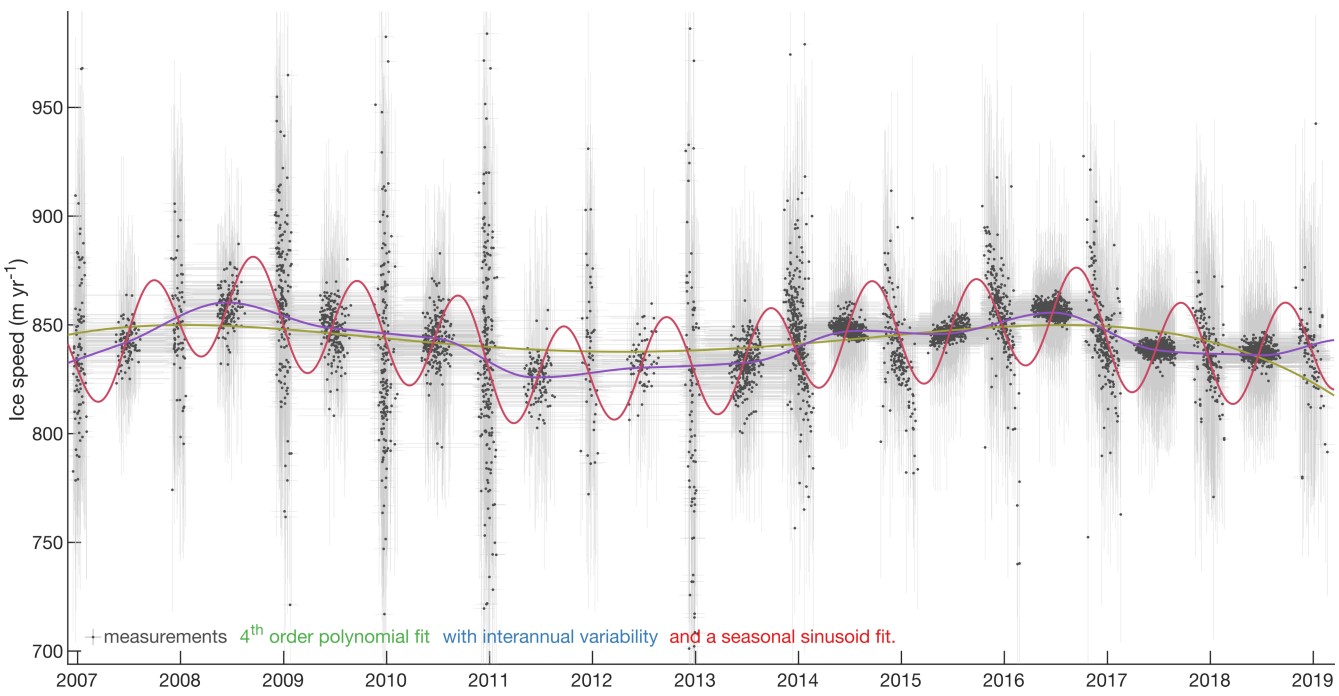

**Figure 4. Velocity time series for an ITS_LIVE pixel.** These 14,208 measurements taken near the grounding line of Byrd Glacier typify ITS_LIVE image pair data, with short-$\Delta t$ measurements providing direct, but noisy observations of velocity variability throughout each summer, while much lower-noise winter estimates can only give insight into the total displacement that occurs during the dark, winter months. Light gray horizontal bars connect the acquisition times of each image pair and vertical bars show $\pm 1\sigma_\mathrm{v}$ uncertainty. Center dates $t_\mathrm{m}$ are shown as dark gray dots for visual clarity.

describing cyclic behavior, and by nature it is constrained to capture the sum total of ice displacement that occurs throughout the year.

After characterizing interannual variability, we subtract it from the velocity time series at the center date of each image pair. The residuals $v_\mathrm{r}$ after removing interannual variability can then be assumed to contain only seasonal variability and noise. To
address blunders, we remove outliers whose absolute value exceeds 2.5 robust standard deviations of $v_\mathrm{r}$ (see Appendix A). The remaining task is to fit a sinusoid to the $v_\mathrm{r}$ time series, but given that each velocity measurement is a single value that represents several weeks to more than a year, we cannot fit a sinusoid directly to the velocity time series or assume that values of $v_\mathrm{r}$ correspond to the image pair center dates $t_\mathrm{m}$. Instead, we operate on the displacements associated with each image pair, taken as the integrals of velocities $v_\mathrm{r}$.
We seek to define seasonal velocity variability $v_\mathrm{s}$ in the form

$$v_\mathrm{s}(t) = A\sin\left(2\pi(\phi + t)\right) + C_0, \tag{1}$$





where $A$ and $\phi$ are the amplitude and phase of the seasonal velocity cycle, respectively, and $t$ represents time in decimal years. By our method, the constant $C_0$ is ideally zero after detrending and removing interannual variability, but we include it here in case some residual offset is present in the $v_r$ time series.

For a more robust least-squares solution, we employ a trigonometric identity to rewrite Eq. 1 as

$$v_s(t) = C_1 \sin(2\pi t) + C_2 \cos(2\pi t) + C_0, \tag{2}$$

which is related to Equation 1 by $A = \sqrt{C_1^2 + C_2^2}$ and $\phi = \mathrm{atan2}(C_2, C_1)$.

Again, we cannot solve Eq. 1 or Eq. 2 directly with image-pair velocities, because they do not represent instantaneous velocity at any known times $t$. Instead, image pair data track displacements, which equate to the integral of velocities between 120 times $t_1$ and $t_2$ when the two images were acquired. Accordingly, we can take the definite integral of Eq. 2 to solve for the seasonal displacement cycle $d_s$ as,

$$d_s = \frac{C_1}{2\pi} \left[ \cos(2\pi t_1) - \cos(2\pi t_2) \right] + \frac{C_2}{2\pi} \left[ \sin(2\pi t_2) - \sin(2\pi t_1) \right] + C_0(t_2 - t_1). \tag{3}$$

We solve for the coefficients of Eq. 3 using a least-squares approach with weights given by $w_d = (\sigma_v \cdot \Delta t)^{-2}$. This type of approach has previously been applied to study Earth deformation (e.g., Hetland et al., 2012), and is similar to an approach that 125 has been used to analyze ice dynamic responses to tidal forcing (Milillo et al., 2017; Minchew et al., 2017).

By employing Eq. 3 to solve for $A$ and $\phi$, we obtain a first approximation of the seasonal variability of ice velocity. However, the possibility remains that the initial estimate of interannual variability may have been partly aliased by seasonal variability due to uneven temporal sampling. Thus, we refine our estimates of interannual and seasonal variability by iterative means.

### 3.3 Iteratively refine interannual and seasonal variability estimates

If our first estimates of $A$ and $\phi$ are correct, then we know that the amount by which seasonal variability aliased each initial estimate of interannual variability is given by

$$v_a = \frac{A}{2\pi \Delta t} \{ \cos \left[ 2\pi(t_2 + \phi) \right] - \cos \left[ 2\pi(t_1 + \phi) \right] \}. \tag{4}$$

After obtaining initial estimates of $A$ and $\phi$, we subtract $v_a$ from the original detrended velocity measurements and repeat the process of calculating interannual and seasonal variability. In most cases, we find that initial estimates of $A$ and $\phi$ are 135 reasonably close to their true values, and that just a few iterations are sufficient to yield accurate final estimates of seasonal amplitude and phase. We explore the number of requisite iterations in Sect. 4.3 and show the effects of iterating in Fig. 5.

Figure 4 shows an example of our method applied to a time series of 14,208 ITS_LIVE image pairs acquired in a single pixel near the grounding line of Byrd Glacier, Antarctica. Because no strong long-term velocity trends are present, the green fourth-order polynomial fit is relatively flat, albeit with a slight downturn at the unconstrained end of the time series. The blue 140 curve of interannual variability follows the multi-year rises and falls of velocity much more closely, and is uncontaminated by seasonal variability on this tenth iteration of the solution. The red curve adds seasonal variability with an amplitude of 23 $\mathrm{m\,yr^{-1}}$ to the interannual variability, and given that it is a least-squares solution, it represents the only solution that minimizes





the misfit between the curve and all available observations. Indeed, while at first glance it may appear visually that the timing of image acquisitions themselves are the only seasonal pattern in the ITS_LIVE data plotted in Fig. 4, close inspection shows a

persistent pattern of summer slowdown in the image pair data that becomes especially clear after the 2013 launch of Landsat 8.

## 4 Sensitivity analysis

To assess uncertainty of our method, we generate synthetic random continuous velocity time series, then artificially sample them using random subsets of image acquisition times from Landsat image pairs that cover Antarctica. We then apply the methods described in Section 3 to determine how accurately seasonal ice dynamics can be measured, and under what conditions.

### 4.1 Synthetic time series generation

Any synthetic velocity time series used for testing should resemble the true nature of ice dynamics in its variability on all timescales. Accordingly, we create realistic interannual variability by matching the distribution of the standard deviations of velocities in each grid cell in the ITS_LIVE annual velocity mosaics from 2013 to 2018. Figure 3 shows a map of interannual variability for the grid cells that contain all six years of data, and a histogram of those values for grid cells that contain at least

100 total image pairs and a minimum mean velocity of 15 m yr$^{-1}$. Within this subset, the median interannual variability is characterized by a standard deviation of 4.2 m yr$^{-1}$.

  To create each synthetic time series of interannual variability, we generate uniformly distributed random values centered about zero, at daily temporal resolution. We apply a first-order low-pass Butterworth filter with a cutoff period of 548 days to each random time series to ensure that no annual cycles are present, and then we discard the first and last 548 values of

each time series to eliminate edge effects of the filter. We then scale the remaining time series such that its standard deviation matches a prescribed level of interannual variability.

  Our handling of interannual variability is an attempt to mimic the observable ways in which Antarctic glaciers speed up and slow down from year to year. Ideally, we would also carry forth in a similar manner for seasonal variability, imposing cyclic behavior that matches the true character and distribution of the types of seasonal variability that exist in nature. However, as

the intent of this paper is to develop the methods that will be necessary to understand where and how ice velocities vary on seasonal timescales, we cannot at present create synthetic seasonal variability distributions to match what truly exists in nature. Instead, to each synthetic time series we add a sinusoid with a period of 365.25 days, a random phase, and a random amplitude between 0 and 100 m yr$^{-1}$.

### 4.2 Synthetic time series sampling

Each synthetic time series is artificially sampled using characteristic acquisition times and error distributions from Antarctic Landsat-derived ITS_LIVE image pairs. In Fig. 3 we see that among grid cells containing at least 100 image pairs, and whose mean velocity is at least 15 m yr$^{-1}$, we can expect a median of 1153 image pairs per grid cell. Accordingly, we use the acquisition times (Fig. 2) and corresponding error estimates (Fig. 3) from random subsets of 1153 image pairs (sampled from





**Table 1. Sensitivity test parameters.** To understand how various factors influence measurement sensitivity, we isolate and vary a number of key characteristics of the synthetic time series and the data analysis method. In the first test, we vary only the number of iterations described in Section 3.3, to determine how many iterations are necessary to achieve a stable solution. We then test the effects of sampling velocity time series with 32 to 10,000 synthetic image pairs. In the remaining tests, we prescribe characteristics of the synthetic velocity time series to understand the influence of interannual variability, the amplitude of the seasonal cycle, and the timing of the seasonal cycle on measurement accuracy.

| test | iterations | $N$ image pairs | interannual rms (m yr$^{-1}$) | seasonal amplitude $A$ (m yr$^{-1}$) | seasonal phase | image times & velocity errors | Figure |
|---|---|---|---|---|---|---|---|
| iterations | 1–15 | 1153 | 4.2 | 0–100 | 0–2$\pi$ | RFD | 5a,b |
| $N$ image pairs | 10 | 32–10,000 | 4.2 | 0–100 | 0–2$\pi$ | RFD | 5c,d |
| interannual rms | 10 | 1153 | 0–3500 | 0–100 | 0–2$\pi$ | RFD | 5e,f |
| seasonal amplitude | 10 | 1153 | 4.2 | 0–500 | 0–2$\pi$ | RFD | 5g,h |
| seasonal phase | 10 | 1153 | 4.2 | 0–100 | 0–2$\pi$ | RFD | 5i,j |
| overall Antarctic | 10 | RFD | RFD | 0–100 | 0–2$\pi$ | RFD | 6 |

RFD indicates *randomized from distribution* of all 1.8 million Antarctic ITS_LIVE image pairs.

all 1.8 million Antarctic image pairs) to artificially sample each synthetic time series. Each synthetic velocity measurement
is calculated from the cumulative sum of daily velocities that occurred between the first and second image in a given pair, but before dividing the total displacement by the time $\Delta t$ between images, a random gaussian value of displacement is added according to the formal error estimates associated with that image pair in the ITS_LIVE dataset. The result is a time series of synthetic measurements that resemble the acquisition times and error characteristics of ITS_LIVE image pairs, but whose underlying continuous velocity signal it represents is known.

**4.3    Seasonal amplitude and phase recovery**

We conducted several tests to determine the accuracy with which we can recover the amplitudes and phases of seasonal cycles in synthetic velocity time series. The parameters of each test are detailed in Table 1.

In the first test, we sought to understand how many iterations are necessary to achieve a stable solution. In this test, we generated 10,000 synthetic velocity time series, each having interannual variability with a standard deviation of 4.2 m yr$^{-1}$.
Sinusoidal seasonal variability was added to each time series, characterized by a random phase and a uniform distribution of amplitudes between 0 and 100 m yr$^{-1}$. Each time series was then synthetically sampled using the acquisition times and error characteristics of 1153 random image pairs in the ITS_LIVE dataset. Following the method described in Sect. 3, we analyzed each time series by assessing interannual variability in the synthetic measurements, removing it, removing outliers, and then obtaining the amplitude and phase of each seasonal cycle. After accounting for any aliasing that would have been caused by the
seasonal amplitude and phase (Sect. 3.3) that were measured in the first iteration, we conducted a second iteration of assessing interannual and seasonal variability. We then followed with a third iteration, and so on, up to 15 iterations. Results in Fig. 5a,b



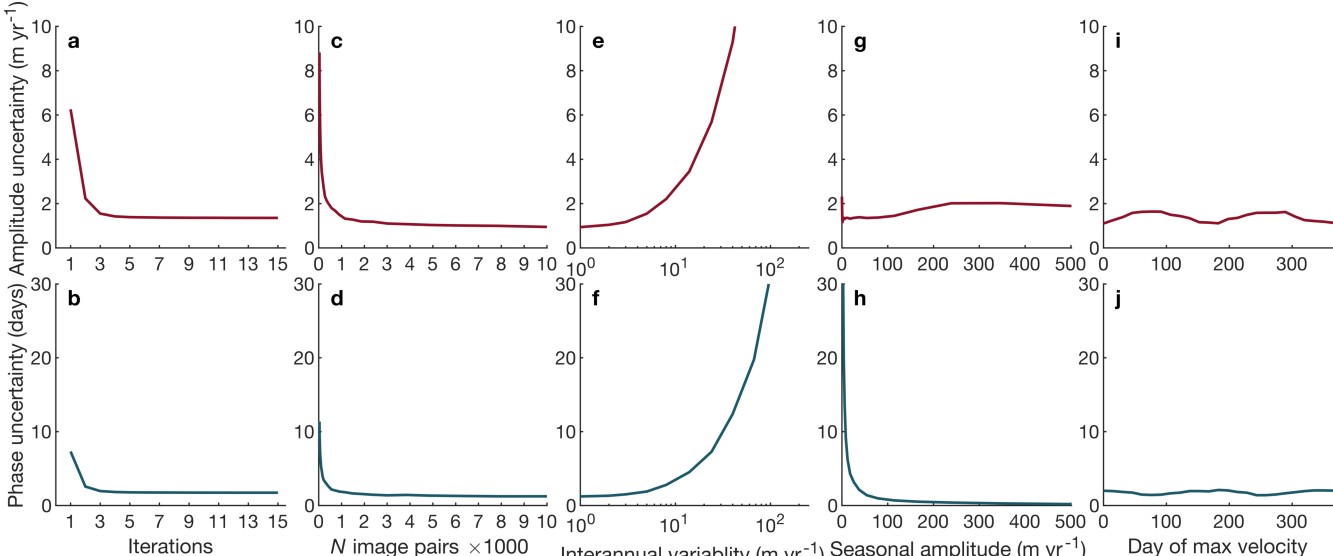

**Figure 5. Sensitivity test results.** The tests outlined in Table 1 indicate that **a–b:** amplitude and phase errors level off after the first few iterations of removing interannual variability and solving for seasonal variability; **c–d:** at least 500 to 1000 image pairs are necessary for the most accurate, stable solutions; **e–f:** strong interannual variability can drastically affect seasonal amplitude and phase detection; **g:** seasonal amplitude measurement uncertainty is independent of the seasonal ice velocity amplitude itself, but **h:** seasonal phase is measured most accurately when the seasonal ice velocity amplitude is strong; and **i–j:** although some faint residual effects of the seasonal bias in sampling persist after 10 iterations, amplitude and phase accuracy are effectively independent of the timing of the seasonal ice velocity variability.

show that solutions tend to converge after the first few iterations, but it is possible that in some situations more iterations could be necessary, depending on sampling and the characteristics of the velocity time series. Accordingly, we use 10 iterations in all of the tests that follow.

We conducted a second test to determine how many image pairs are necessary to detect seasonal cycles in a velocity time series. We found that with just 32 image pairs, we were able to recover phase with sufficient accuracy to correctly identify the season of maximum velocity (Fig. 5c,d). We also found that performance improves dramatically with increasing image pairs, until errors in phase and amplitude approach their asymptotes between 500 and 1000 image pairs. Beyond that, the number of image pairs used in the analysis has a negligible impact on the accuracy of seasonal signal detection.

In a third test, we found that one factor more than any other threatens the accuracy of seasonal variability detection. Given a velocity time series sampled by 1153 image pairs, seasonal amplitude error increases approximately linearly with the level of background interannual variability (Fig. 5e,f). This is because despite our attempts to account for interannual variability, it is inevitable in a time series of finite length that some residual variability will influence the least-squares fit of the seasonal cycle. Nonetheless, if we consider ±45 days of phase uncertainty to be a threshold that indicates accurate detection of the season of maximum velocity, our method performs adequately in the presence of interannual variability with a standard deviation

exceeding 100 m yr$^{-1}$. Further analysis suggests that for any level of interannual variability, the amplitude of the seasonal





cycle must be at least one third of the standard deviation of interannual variability to reliably be detected (see Sect. A1). We note, however, that because we have used the simple metric of the standard deviation of velocity to describe all forms of interannual variability, it is likely that our method will perform better than these error estimates suggest wherever interannual
variability is dominated by a long-term trend.

The final four panels of Fig. 5 provide valuable insights into the capabilities and sensitivities of our seasonal detection method. Most notably, that for a constant level of background interannual variability, seasonal amplitude errors are independent of the amplitude of the seasonal cycle. However, phase detection benefits with increasing seasonal amplitude. This should not be surprising, as a signal must not only exist, but also be sufficiently strong for its phase to be accurately measured. The last two
panels of Fig. 5 show that for all practical purposes our seasonal amplitude and phase estimates can be considered insensitive to the timing of the underlying seasonal signal, despite having no images during winter.

Thus far we have determined that 10 iterations are more than sufficient for the model of seasonal variability to converge. We have also found that our measurements can be considered agnostic to the timing of ice velocity variability and to the timing of satellite image acquisitions, when applied to at least 500 to 1000 image pairs. With this understanding, we now apply our
method to 100,000 synthetic time series that typify the interannual ice velocity variability and satellite image acquisitions that have been measured across the Antarctic continent as a whole. Interannual variability in the synthetic time series was randomly sampled from the distribution shown in Fig. 3d and a uniform distribution of seasonal amplitudes from 0 to 100 m yr$^{-1}$ were added to the interannual variability. Each synthetic ice velocity time series was observed with a number of synthetic image pairs taken randomly from the distribution of values in Fig. 3c that exceed 500 image pairs, and displacement errors randomly
sampled from the distribution in Fig. 3e were added to each synthetic measurement before dividing by the times $\Delta t$ that separate each image pair. The results of this test, shown in Fig. 6, indicate that with the Landsat images that have been acquired to date, we should be able to measure seasonal amplitudes within $\pm 1.4$ m yr$^{-1}$ and phase within about $\pm 2$ days.

## 5  Validation with in situ GPS observations

To confirm that we can extract seasonal ice velocities from feature-tracking data, we compare an *in situ* GPS position time
series against results obtained by applying our method to ITS_LIVE velocities that were generated from 15 m resolution optical Landsat 7 and 8 imagery. We focus on Russell Glacier in Greenland, where GPS data provide more than a decade of observations and where seasonal velocity variability has previously been reported Van de Wal et al. (2015). We use the GPS-derived positions from the PROMICE (Van As, 2011) KAN_L station (64.4822°N, 49.5358°W, 530 masl; see Appendix B). Although we ultimately aim to characterize Antarctic seasonality, we use this record from Greenland because we are not
aware of any such decade-long GPS records in Antarctica that offer winter position data and capture seasonal cycles of glacier movement that can be used for validation. The ITS_LIVE data covering Greenland are identical to the Antarctic image pair data, so we directly apply the methods discussed above without any adjustments. Likewise, our methods could just as easily be applied to ITS_LIVE data covering Alaska, the Canadian Arctic, Svalbard, the Russian Arctic, Patagonia, or High Mountain Asia.

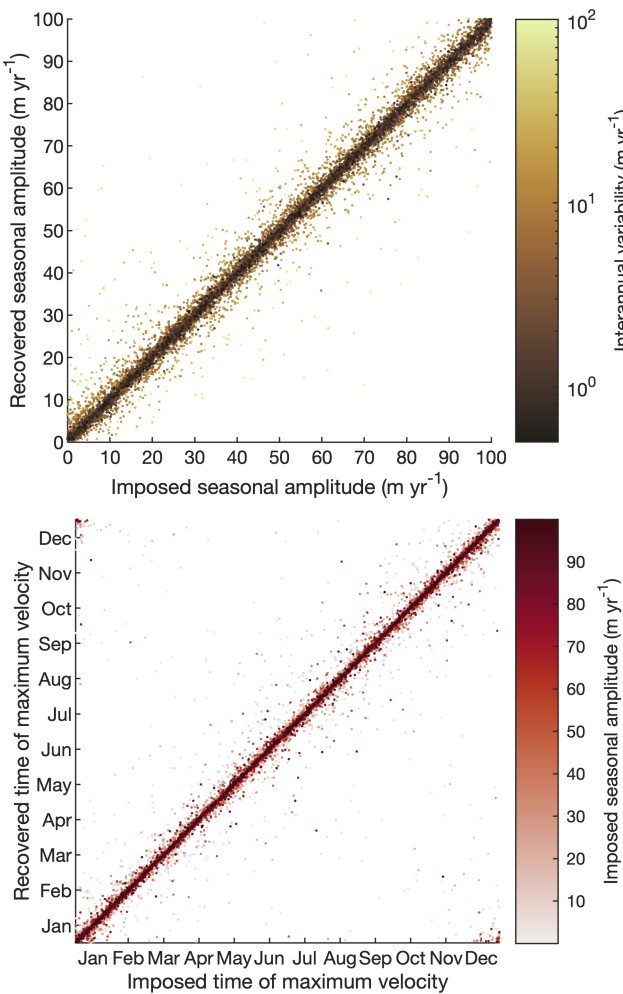

**Figure 6. Pan-Antarctic seasonal signal recovery.** Following the parameters of the final test in Table 1, we examine the performance of our method using the complete distributions of Antarctic Landsat image acquisition times, ITS_LIVE velocity errors, and measured Antarctic interannual ice velocity variability. Among the results where 500 or more image pairs were used to analyze the time series, amplitude uncertainty was found to be $1.4 \, \mathrm{m \, yr^{-1}}$ and phase uncertainty is 2 days. For visual clarity, only 10,000 randomly selected results are shown of the 100,000 tests we performed for overall Antarctic parameter distributions.

Figure 7a,b shows the linearly detrended GPS position data, along with a model fit that represents the sum of interannual variability and a seasonal sinusoid fit to the position data (see Appendix B). From the GPS data, we obtain velocity time series by taking the time derivative of the position fits, which are shown in Fig. 7c,d. As we expect when taking the derivative of a sinusoid, the peaks in velocity occur 91 days before peaks in position, and velocity amplitudes are $2\pi \, \mathrm{yr^{-1}}$ times the amplitudes of the seasonal position fits.



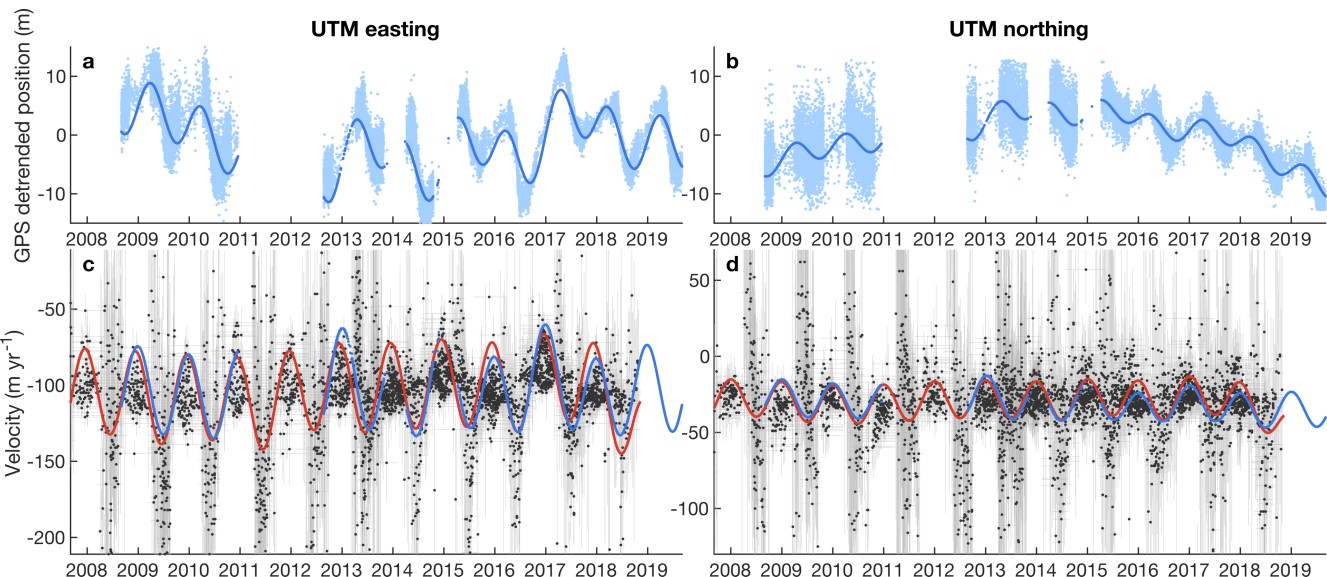

**Figure 7. GPS validation at Russell Glacier, Greenland.** Hourly time series of **a–b:** detrended positions of a GPS receiver are shown as raw data in light blue and the sum of interannual variability and a sinusoid fit in dark blue. The amplitudes of positional seasonal variability are only 4.5 m in the easting ($x$) direction and 1.8 m in the northing ($y$) direction, yet these minor seasonal displacements are easily detected by applying the methods described in Sect. 3 to ITS_LIVE Landsat image pair velocities. **c–d:** Velocity measurements from 5189 ITS_LIVE image pairs are plotted in gray with our fit to the measurements in red. The blue velocity curves are simply the time derivatives of the corresponding GPS position curves plotted in panels a and b. The level of agreement between the two independent measurements is detailed in Table 2.

We compare the GPS velocity time series to velocities obtained from 5189 ITS_LIVE image pairs in the pixel closest to the fixed median location of the GPS receiver throughout the course of its data collection. Key findings are listed in Table 2. Secular mean velocities in the $x$ and $y$ directions measured by the two independent datasets agree on the order of 1 m yr$^{-1}$. Seasonal velocity amplitudes also show agreement on the order of 1 m yr$^{-1}$, as we expect on this glacier where interannual variability has a standard deviation of 3.6 m yr$^{-1}$ in the $x$ direction and 3.4 m yr$^{-1}$ in the $y$ direction (see Appendix A2).

The largest disagreement in Table 2 lies in the phase of the $x$ direction, which at 14 days is only about 4% of a year. We note further that disagreements listed in Table 2 do not necessarily reflect errors in the ITS_LIVE data or the methods we have employed to process it. Rather, disagreements may simply reflect that the GPS receiver did not record data from late 2010 to mid 2012, and no GPS data were recorded during two of the winters that followed. Meanwhile, we use ITS_LIVE data from images that were collected more than a year before the start of the GPS record, but image pair data have not yet been processed corresponding to the final months of the GPS record. In addition to these differences in timing of data collection, we also note that the GPS record represents a Lagrangian measurement of ice velocity, whereas the ITS_LIVE data record an Eulerian measurement at a pixel that the GPS receiver passed through only once in its decade-long life. Nonetheless, agreement between



**Table 2. ITS_LIVE analysis compared to GPS.** The secular mean velocities and $x$ and $y$ components of the seasonal characteristics of the glacier velocity time series shown in Fig. 7. The acquisition dates of the two independent measurements differ slightly, yet agreement is within a few percent by every measure. Uncertainty estimates are described in Appendix A2.

|  | GPS | ITS_LIVE | difference |
|---|---|---|---|
| secular mean $\bar{v}_x$ | -102.4 | -104.0 | -1.6 m yr$^{-1}$ |
| $v_{\mathrm{s},x}$ amplitude | 28.5 | 28.8±0.9 | 0.3 m yr$^{-1}$ |
| day of max $v_{\mathrm{s},x}$ | Dec. 27 | Dec. 13±5 | -14 days |
| secular mean $\bar{v}_y$ | -30.6 | -29.5 | 1.2 m yr$^{-1}$ |
| $v_{\mathrm{s},y}$ amplitude | 11.5 | 12.4±0.9 | 0.9 m yr$^{-1}$ |
| day of max $v_{\mathrm{s},y}$ | Jan. 3 | Jan. 1±11 | -2 days |

the GPS solutions and our method of image pair data analysis is remarkable, and lends credence to this method as a robust means of measuring seasonal variability in ice velocity.

## 6 Discussion

The method we present in this paper requires a multi-year record with at least several hundred image pairs to confidently identify the amplitude and phase of seasonal variability. Some key regions of interest, such as Totten Glacier in Antarctica, do not yet offer enough cloud-free images to meet this threshold, so a few more years of data collection may be required before our methods can be successfully implemented there. In addition, it may be difficult or impossible by our methods to detect seasonality in places of interest such as Pine Island and Thwaites glaciers, which are currently undergoing dramatic interannual change that could confound our measurements of seasonal variability.

Nonetheless, Fig. 7 illustrates the sensitivity of our method to minuscule variations in glacier flow when conditions are favorable. The vertical scale of the upper panels spans just ±15 m; yet, by our method of analyzing 15 m resolution Landsat Band 8 images, we are able to detect minor nuances in position that occur entirely within this range. We know of no other sensor or dataset that can offer such insights into these kinds of subtle variabilities of ice flow that have occurred over the past few decades.

The most significant limitation of the method we present may lie in our approximation of seasonal variability as a sinusoid. Where seasonal dynamic variability has previously been documented, it has been found that in some cases a sawtooth function or other higher-order fits might match seasonal variability more closely than a sinusoid (Moon et al., 2014; Van de Wal et al., 2015; Vijay et al., 2019). In particular, although events such as springtime calving or summer melt may occur on an annual cycle, a glacier's complete response to them may only last for only a few days (Schoof, 2010). To approximate such events as smoothly varying sinusoids will underestimate the magnitude of any brief glacier acceleration, while potentially giving a false sense that the ice responds to an impulse event continually throughout the entire year.



Despite the tendency of a sinusoid to oversimplify complex time series, we contend that no other description of seasonal variability is as elegant or robust over decadal timescales, and that understanding seasonal ice dynamics must begin with a zeroth order description of the amplitude and phase of ice velocity variability. While it is true that a glacier can accelerate in response to a transient event and return to an equilibrium velocity within just a few days (Stevens et al., 2015; Andrews et al., 2014), in the climatological sense, nature does not consistently time such events as calving or increases in basal water pressure with any greater precision than the method we have presented to detect them.

In most cases, a sinusoid will likely capture the majority of velocity variance throughout the year, and represent the fundamental mode of subannual variability in ice velocity. The approach we present conserves all ice displacement that occurs throughout the year, and the simple two-term explanation of amplitude and phase provides a robust description of seasonality that is less prone to error than higher-order fits such as three-term sawtooth functions. By providing a simple measure of amplitude and phase, we offer a straightforward method to compare how neighboring glaciers respond to a common seasonal forcing or investigate how dynamic signals propagate upstream or downstream in a given glacier.

In Sect. 5 we applied our method to ITS_LIVE velocity measurements and compared against GPS position data, with the intention that the *in situ* GPS station would provide a reliable ground-truth reference. However, although the two independent measurements show close agreement whenever GPS data were logged, the receiver's harsh polar environment has led to several long gaps during which no GPS position data were acquired. This suggests that as a means of measuring ice dynamic climatology, our method might not only meet, but exceed the performance of the *in situ* GPS receiver while providing insights into dynamic behavior as far back as the mid 1980s. The particular ITS_LIVE grid cell in Greenland that we analyzed in this paper is hardly unique in its potential to provide historical context, as decades-long velocity records exist in most of the $240{\times}240\,\mathrm{m}$ pixels which cover nearly all of the world's ice.

The methods presented in this paper have focused primarily on optical satellite data because no other type of sensor provides such a long record of ice velocity. As more radar data become available, particularly since the launch of Sentinel 1a/b, the problem of missing winter data will be eliminated, but the methods presented in this paper will still hold. When an abundance of feature-tracked velocities from radar become available, Eq. 3 may then be easily modified to include additional terms to simultaneously solve for cyclic velocity variability on tidal timescales as well as seasonal variability.

## 7   Conclusions

Given a relatively continuous time series of at least 500 to 1000 image pairs, our method can extract the amplitude of seasonal variability with a precision on the order of about $1\,\mathrm{m\,yr^{-1}}$, provided the level of background interannual variability does not overwhelm the overall signal. We find that if the amplitude of seasonal variability is at least a third of the standard deviation of interannual variability, the method we describe can reliably detect the season of maximum ice velocity. Ability to detect seasonal amplitudes is independent of the amplitude and phase of the seasonal velocity variability, but phase accuracy benefits with increasing amplitude of seasonal variability.





With the method we describe, we may begin to map seasonal ice dynamic variability on a global scale, in a consistent and meaningful manner; and by providing a method that can be employed independently in the two dimensions of Cartesian coordinates, we hope to gain a fully three dimensional understanding of how dynamic signals propagate through the world's ice.

*Code and data availability.* Data analysis in this paper relied upon Antarctic Mapping Tools for MATLAB (Greene et al., 2017b) and the Climate Data Toolbox for MATLAB (Greene et al., 2019). All data analyzed in this paper and code necessary to generate the figures are included as a supplement to the paper, or are available online at http://www.github.com/chadagreene. PROMICE data is freely available at http://www.promice.dk. ITS_LIVE global image-pair velocity data is freely available at its-live.jpl.nasa.gov.

## Appendix A:  Additional uncertainty analysis

Phase and amplitude uncertainty in Sect. 4.3 were calculated from the differences between imposed and recovered values. Differences in phase were wrapped such that their absolute values did not exceed 182.62 days. During this work, we found that in some cases, simple standard deviations of differences could be strongly influenced by a few egregious outliers, and as a result, standard deviations of differences (and similarly, root-mean-square errors) did not reflect the true gaussian distributions of errors. So for a more robust and meaningful measure of error distributions, we define $\sigma$ as 1.4826 times the median of the 325   absolute value of differences.

### A1   Seasonal uncertainty from interannual variability

In addition to the tests described in Sect. 4.3, we analyzed 120,000 synthetic time series to better understand the scenarios in which interannual variability may confound our ability to detect seasonal cycles in an ice velocity time series. For every unique combination of 31 values of seasonal amplitudes from 0 to 1000 m yr$^{-1}$ and 26 values of interannual variability from 0 to 3500 330   m yr$^{-1}$, we generated 150 synthetic time series, sampled them with 1153 synthetic image pairs, and attempted to recover the seasonal cycles we imposed. A striking relationship emerged, shown in Fig. A1, in which we see a strong demarcation between a zone of accurate phase detection and a zone where phase cannot be determined with any confidence.

Letting phase uncertainty of 45 days be the threshold indicating whether the season of maximum ice velocity can be accurately determined, in Fig. A1 we see that above the noise floor of about 1 m yr$^{-1}$ seasonal amplitude, the 45 day phase 335   uncertainty contour marks an approximately linear relationship between interannual variability and seasonal amplitude. Indeed, the dashed blue line that nearly coincides with the 45 day phase uncertainty contour shows the simple slope whereby seasonal amplitude is one third of the standard deviation of interannual variability, with zero offset from the origin. This tells us that there is a quite simple relationship between the amplitude of a seasonal cycle, the level of background interannual variability, and our ability to detect phase—As long as the seasonal amplitude is above the noise floor of about 1 m yr$^{-1}$ and

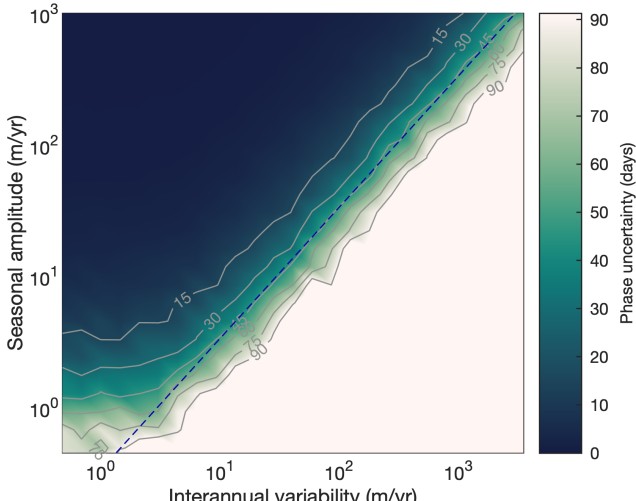

**Figure A1. Interannual variability and seasonal signal detection.** The dark green region of this plot indicates scenarios in which phase can be accurately detected with 1153 image pairs. When seasonal amplitudes are above the noise floor of about $1\,\text{m}\,\text{yr}^{-1}$, the amplitude of seasonal variability must be at least a third of the standard deviation of interannual variability to be detected, and this relationship is marked by a dashed blue line. The linear relationship coincides with the 45 day phase uncertainty contour that determines whether the season of maximum ice velocity can be accurately measured.

interannual variability does not exceed three times the amplitude of the seasonal cycle, we can accurately detect the seasonal cycle.

## A2   Uncertainty estimates for ITS_LIVE/GPS comparison

The values of seasonal amplitude and phase uncertainty listed in Table 2 were each calculated as the robust standard deviations of amplitude and phase errors from 5000 synthetic time series sampled by 5189 image pairs. In the $x$ direction, ITS_LIVE

image pairs indicate an interannual variability of $3.85\,\text{m}\,\text{yr}^{-1}$ in our pixel of interest, with a seasonal amplitude of $28.8\,\text{m}\,\text{yr}^{-1}$ and a maximum velocity occurring around December 13th of each year. We generated 5000 synthetic time series matching these criteria, and were able to recover the $28.8\,\text{m}\,\text{yr}^{-1}$ seasonal amplitude within $\sigma=0.87\,\text{m}\,\text{yr}^{-1}$ and phase within $\sigma=4.7$ days. Similarly for the $y$ direction, from 5000 synthetic time series with an interannual variability of $3.38\,\text{m}\,\text{yr}^{-1}$, seasonal amplitude of $14.4\,\text{m}\,\text{yr}^{-1}$, and maximum velocity occurring on January 1 of each year, we recover seasonal amplitudes within

$\sigma=0.91\,\text{m}\,\text{yr}^{-1}$ and phase within $\sigma=10.7$ days.

## Appendix B:  GPS processing

We use hourly position data from the PROMICE KAN_L GPS station on Russell Glacier (Van As, 2011). The KAN stations utilize the Trimble SAF270-G antenna with a single L1 frequency to minimize power usage. L1 signals have previously been



used in studies of short-term ice motion with success in the Russell Glacier region (e.g., Van de Wal et al., 2015). Positions are

recorded hourly during the spring/summer observation period and daily over winter. In addition to position data, PROMICE reports information about horizontal dilution of precision (HDOP) and station tilt data; both of which are used in addition to position data to perform post-processing quality control. HDOP provides information on uncertainties related to satellite geometry, and we discard positions for which the HDOP exceeds a value of 5. We manually remove offsets of a few meters that occurred during four site visits on 28 April 2015, 16 July 2016, 1 Sept 2017, and 28 Aug 2018. We then remove any

outliers, which we define as all points whose detrended $x$ or $y$ positions lie more than 2.5 robust standard deviations (see Appendix A1) away from zero. After detrending the position data we remove interannual variability following the spline-fitting method described in Sect. 3.1, then fit sinusoids to the residual $x$ and $y$ position time series using the `sinefit` function in MATLAB (Greene et al., 2019). The detrended raw position data are shown in Fig. 7 along with a model fit, which is taken as the sum of interannual and seasonal variability.

*Author contributions.* CAG conceived of and carried out the study with guidance and technical assistance from ASG. LCA post-processed and interpreted the GPS data. CAG wrote the manuscript with input from ASG and LCA.

*Competing interests.* The authors declare that they have no competing interests.

*Acknowledgements.* The authors were supported by the NASA Postdoctoral Program, the NASA Cryosphere program, and the NASA MEa-SUREs program. Data from the Programme for Monitoring of the Greenland Ice Sheet (PROMICE) and the Greenland Analogue Project

(GAP) were provided by the Geological Survey of Denmark and Greenland (GEUS) at http://www.promice.dk. The research was conducted at the Jet Propulsion Laboratory, California Institute of Technology under contract with the National Aeronautics and Space Administration.



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
