# Peer review of "Detecting seasonal ice dynamics in satellite images"

_The Cryosphere, 2020_

## Referee Comment (RC1) · Anonymous Referee #1 · 23 Jun 2020

The authors have proposed a method to characterize the magnitude and timing of seasonal glacier ice velocity signals by fitting the best possible sinusoid to the velocity data obtained from optical remote sensing. It is well known that the optical data has obvious data gaps in polar regions during winters and is affected by cloud cover. This method is proposed to resolve seasonal velocity variations from such a dataset, but needs a large number of (>1000) multi-year velocity observations. The manuscript is well written, but I have a couple of points that may be useful to further improve this work. My major concern is about the applicability of this method to regions other than polar areas.

Major comments:

P5. I did not understand how observations over finite integration times make it difficult

to resolve seasonal variability in case of repeat SAR imagery. For instance, Sentinel-1 SAR data is available throughout the year with a 6-day temporal cycle and a number of studies have resolved seasonality using SAR data (e.g. Sentinel-1, TanDEM-X). I think the focus of this paper should be optical data, its limitations during polar winters and cloud cover and how your method can still resolve seasonality using optical data.

P30-35. The authors should highlight these significant gaps which still limit our understanding of ice dynamics change on different time scales. I agree that a number of studies on the seasonal ice dynamics of glaciers in Arctic, Antarctic and other glaciated regions are available in bits and pieces, but they do provide a great degree of evidence that help us understand the physical processes which govern the ice dynamics on different time scales. I can't imagine how a consistent global mapping of seasonal ice dynamics looks like, which the authors have pointed to and how, if accomplished, they better our understanding. It is also not clear why such an approach relies on optical imagery, even though we have a year-around consistent and global SAR imagery by missions like Sentinel-1. These points should be addressed in the Introduction to better form a basis or need for this study.

Figure 7. Nice figure. But when I compared this with Figure 4, I drew a couple of points that need to be addressed. ITS_LIVE velocity data for Russel Glacier in Greenland is much more dense and appears to be well distributed around the year as compared to Byrd Glacier, Antarctica. I wonder how such a large number of wintertime velocities are available in Greenland using optical data. Are they averaged for the entire polar wintertime? I expect that this dense and well distributed velocity data is the main reason why you have a great sinusoidal fit here, isn't it? Because ideally the method should resolve the missing velocities in winters using the data points for the rest of the year. By the way, it appears to me that ITS_LIVE observations in case of Russel Glacier are already resolving seasonal variations without applying your proposed method. An example, where velocity data is sparse like for Byrd Glacier, would be much more convincing how well your proposed method can resolve seasonality. Russel Glacier is

the best case scenario (P270). An example from any mountain glaciers from Alaska, European Alps and high Asia would enhance the applicability of the proposed method. The optical data is available around the year in these regions and is only affected by cloud cover. At present the method is proposed to work in these regions, but the potential challenges are not highlighted.

Minor comments:

P5,25 or elsewhere. Instead of using "accelerations", I would recommend to use "velocity variations" because both acceleration and deceleration are governed by physical processes. P25. Can your approach resolve velocity changes during such short time-scales? If yes, you should highlight in the paper what additional information is required in your method in order to retrieve such signals. If not, there is no need to include it in the Introduction.

Figure 1. Example-1 shows a hypothetical scenario, especially the velocity time series shown in blue. We have plenty of SAR data and derived ice velocities for the polar glaciers. What about showing a real case here?

P45: satellite image pairs » optical remote sensing image pairs

P85: It is not clear how the weights are assigned here. Are these based on residuals?

Figure 4: The colors (blue and green) in the legend and fits are inconsistent.

P105. "Instead, we operate on the displacements associated with each image pair, taken as the integrals of velocities" should be shown as a different figure as this is an important step of the paper. It would be better to see how various displacement estimates at different epochs ranging from days to years are prepared for any sinusoidal fit.

P130. I recommend the authors to make a relationship between Va and Vs here. In other words, the authors should derive equation 4 from above equations or make a relationship between them. What is the goal here? Minimizing the Va?

P145. What about showing an additional inset figure here, which zooms in velocity variability for a particular year with X-axis showing weeks or months of that year.

P165. I didn't quite understand why a synthetic seasonal velocity signal can't be created that depicts nature. I would rather insist the authors to do the same or make use of SAR-based year-around velocity observations to create one (e.g. NASA MeASUREs monthly mean ice velocity mosaics by Ian Joughin). Also shouldn't the goal of your study be to resolve seasonality from a number of observations spaced in time?

P230. reported in van de Wal et al., (2015)

---

## Referee Comment (RC2) · Anonymous Referee #2 · 29 Jun 2020

The authors present a workflow to fit a sinusoidal function to a data set of clustered velocity estimates on ice sheets and outlet glaciers. The work is well written, and the authors clearly identify the need to extract more concise information of this vast collection of Eath observation data. The steps taken by the authors are explained, but in general there is a tendency to highlight the strong points of the methodology in their argumentation. Being a methodology paper, there might be a reason to keep this presentation brief, but it might be more than worthwhile to emphasis points of improvement and why certain decisions are taken.

My main concern with this work stems from the property that the authors define seasonal variation as a cycle. In this way the reader is pushed into a certain narrative, which limits how to approach this issue. The authors are correct about the sinusoidal

variation of the forcing (the sun and the seasons), but this does not mean the ice velocity has the same reaction. Personally, I see the seasonal variation more as a perturbation, to which there is a reaction time/response, a peak and fade out/reorganization. Thus a perturbation (including a sinus function, but also a lot of other responses) occurs every year, due to surface melt run-off, but the time span does not need to extend towards a whole year, as is assumed here. If we look at other studies short spikes are clearly visible (Kjeldsen et al. 2017, 10.1002/2017GL074081; Derkacheva et al. 2020; 10.3390/rs12121935), or in the dynamics of a surging icecap (Dunse et al. 2015, 10.5194/tc-9-197-2015) where a step function is seen, that is initiated by meltwater perturbation. So I miss a discussion on how good a sinus-function is as a model. There is only testing of how good the observations meets the model description, and not how good the model fits the observation. Putting everything on "background interannual variability" is a bit easy.

Another question arising is the wording of climatological velocity, I am not able to figure out what the authors mean with this term. This directly also brings me to a second point on the sinus fitting, as it is treated as a cyclic function similar to (Menchew et al., 2017, 10.1002/2016JF003971). They look at a tidal time span, where the forces are highly repetitive in magnitude and phase. However, if this is the case for seasonal glacier velocities is not so clear, as the amplitude of glacier velocity seem to correlate with surface mass balance. This has been observed with GPS in Greenland (van de Wal et al. 2015, 10.5194/tc-9-603-2015) or on Nordenskiöldbreen, Svalbard (van Pelt et al. 2018 10.1029/2018GL077252). But the sinus function of the authors does not take the change in amplitude, from year to year into account. However, this (to me) would be a climatological velocity (if I had to guess what the authors mean).

Other influencing phenomena, like the ocean/front position have similar seasonal amplitude change (Kehrl et al. 2016, 10.1002/2016JF004133). By putting all these into a cyclic function, the signals of phase and amplitude might smooth out. In connection to this, at high latitudes, the coverage is concentrated towards the summer season.

Hence, how do short term perturbation propagate into the velocity estimation? From the synthetic test the methodology can be "considered agnostic", but this is true for reconstruction purposes of a sinusoidal function. It is also not clear where the authors are after, the onset of speed-up, or "identify the seasonal maximum velocity"? Other studies/data/methods are able to find the timing of such speed-up events (Altena et al. 2017 10.3389/feart.2017.00053, Vijay et al. 2019, 10.1029/2018GL081503), though not as precise or automatic as presented here, but are less constrained. So, there are some issues on the amplitude, but also on the phase. The argument of the authors for using a sinus, as it is "elegant" is a bit weak in my opinion. It would very much strengthen the manuscript, if these influencing effects/considerations are highlighted, as it gives handholds on the way forward.

The authors have formulated their estimation procedure by decoupling the x- and y-velocities. Is there a certain reason for this? I can imagine it can be beneficial, to include co-variance functions, so outliers in one dimension are also excluded in the other. In addition, given these phase angles are estimated independently, do the authors see a difference between both axis. If so, this would imply a change in flow direction, if not what would that mean? Also, why did the authors do filtering (using the MAD), and not do robust least squares, or at least use such procedure in the estimation? Neither is it clear to me why several iterations are applied, see (https://ccrma.stanford.edu/~jos/filters/Sum_Sinusoids_Same_Frequency.html), or is the estimation not restricted to a yearly cycle? Or is the iteration not done on the residuals?

The authors run tests on synthetic data, by imposing corruption to individual velocity estimates. This noise is done on an individual basis, which is partly due to measurement noise. But there is also dependent noise, as displacement estimates are derived for pairs of images. Hence, when one image is corrupted for some reason, there is a high probability it propagates to all displacements it is part of. However, this issue is not included into the analysis, though of importance (and due to the synthetic nature,

is possible to generate). This would give more insights then the 32 velocity estimates, stated now.

minor comments:

In general the manuscript is well written, the authors write in their mother tongue, so concerning this issue I am not able to do any better. But for a global audience the wording is sometimes a bit hard; I have learned quite a lot of new words. For sake of easy reading, and not having to go back and forth to a dictionary, please consider changing words a bit. Think of, "unwieldy" or "egregious"

I have tried to understand from the text what is done, and also looked in the code to be able to zoom into the plots/data. But the provided code and plotting does not work, as some functions ("itslive_tsplot" or "itslive_seasonal_deets") are absent.

title: be a bit more specific, maybe change to "Reconstructing seasonal oscillations" also include "glacier ice"

6: "dark polar winters" > "at high latitudes"

8: "climatological average winter velocities" what is meant here?

15: "sufficient quantity of data" is this due to quantity of data, the consistency of campaigns/ monitoring programs or simple availability of large computing power. Or is it opening up of the archives, making historical flow estimation possible (Cheng et al. 2019 10.5194/isprs-archives-XLII-2-W13-1735-2019).

18: "all of the world's ice" large bodies of glacial ice

20: "upended glaciology" Remote sensing is able to get geometric information about the (sub)surface, and is of great aid. To some extent this is a game changer, but it might be fair to also give credit to automatic weather stations, or put it into persective. This have been other great advancement in glaciology, think for example of (Ohmura et al., 1992; Oerlemans et al., 1998).

31: There is also an large collection of studies at the intermediate timescale, which is left out here, dealing with surges. For sake of completeness, this might be included.

37: "the logistical challenge of" what is meant here?

39: "robust extraction" using a robust pre-processing technique, is different from a robust methodology. Given its stiffness (non adaptive) towards one model (a sinusoidal) this might not be a correct formulation. It might be "precise"...?

40: "primarily focused on Antarctica" maybe better: on the ice sheets and their outlets ....?

53: "by feature tracking" add: over longer time spans

55: "the true magnitude" change to "a" instead of "true" or "a well fit"

56: I miss another possibility here, which is common practice in inSAR displacement estimation, being inversion (e.g.: Bontemps et al. 2018, 10.1016/j.rse.2018.02.023, Li et al. 2020, 10.1016/j.rse.2020.111695). This does not make it necessary to work with average time stamps or a model, and can resolves to very small time steps.

64: " first or second image" first and second? maybe be more clear or use "master-slave" "chip-search space" etc. 100: "most robust means" why is this robust, where do you get the reliability from? fig4: maybe it is good to note, why there are two groups of points, as one is +- a year and the other at short time intervals in summer.

btw: the purple line nicely follows the annual velocity clusters!

105: "exceeds 2.5 robust standard deviations" change to median absolute difference (MAD). this is a typical measure.

151: I don’t think this is sensitivity, but more an analysis to get an idea how good the recovery is. As the model is corrupted with noise and then an attempt is made to reconstruct the model. If I understand correctly.

181: "recover", change to we "are able to describe the variation by seasonal cycles" or alike.

258: " is remarkable" subjective wording, please change

258: "robust", precise/accurate might fit better

267: "minuscule variations" subjective wording

283: " in the climatological sense, nature does not consistently time such events as calving or increases in basal water pressure with any greater precision than the method we have presented to detect them". What is meant here?

285: "In most cases, a sinusoidal will likely capture the majority of velocity variance throughout the year, and represent the fundamental mode of subannual variability in ice velocity." Please justify this claim, as this is the corner stone which the whole study is build upon.

291+: It seems the authors put all the misfits of the sinus model on the inaccuracies of the GPS measurements, while this sensor measures all kinds of physics decadal, annual, daily, ...

305: "our method can extract" please add "..by describing ice flow as an oscillation ..." in some way

309: "independent of the amplitude and phase of the seasonal velocity variability", this is not convincingly given

313: "fully three dimensional understanding" what is meant here?

322: "egregious outliers" egregious=outliers

---

## Author Comment (AC1) · 12 Aug 2020

**Reviewer 1**

**General comments:**

The authors have proposed a method to characterize the magnitude and timing of seasonal glacier ice velocity signals by fitting the best possible sinusoid to the velocity data obtained from optical remote sensing. It is well known that the optical data has obvious data gaps in polar regions during winters and is affected by cloud cover. This method is proposed to resolve seasonal velocity variations from such a dataset, but needs a large number of (>1000) multi-year velocity observations. The manuscript is well written, but I have a couple of points that may be useful to further improve this work. My major concern is about the applicability of this method to regions other than polar areas.

**Major comments:**

**P5. I did not understand how observations over finite integration times make it difficult to resolve seasonal variability in case of repeat SAR imagery...**

The sentence in question is from the abstract, and it previously read:

The task of generating continuous ice velocity time series that resolve seasonal variability is made difficult by the finite integration time over which ice velocities are measured...

We have clarified the wording to more effectively convey that measurements of the total displacement that occurs over several months to a year will offer no direct insight into velocity variability that occurs within those months. The section now reads:

The task of generating continuous ice velocity time series that resolve seasonal variability is made difficult by a spotty satellite record that contains no optical observations during dark, polar winters. Furthermore, velocities obtained by feature tracking are marked by high noise when image pairs are separated by short time intervals, and contain no direct insights into variability that occurs between images separated by long time intervals.

**...For instance, Sentinel-1SAR data is available throughout the year with a 6-day temporal cycle and a number of studies have resolved seasonality using SAR data (e.g. Sentinel-1, TanDEM-X)....**

The 6-day Sentinel 1 repeat pattern does not reliably translate to continuous global coverage. Bandwidth limitations have resulted in minimal Sentinel coverage over Antarctica. Elsewhere, such as in southwest Greenland, SAR feature tracking struggles to correlate features between repeat images.

Even in the presence of perfect data coverage, short, 6-day integration times present their own challenges for feature tracking. Namely, the 1-10 m displacement uncertainty achieved with Sentinel 1 SAR image pairs is independent of image temporal separation. This translates to a velocity uncertainty on the order of 100 m/yr or more for an image pair separated by 6 days,

meaning that any seasonal signals smaller than that would likely be difficult to distinguish from the scatter of the measurement noise. For example, here is a synthetic case of two years of perfect data coverage, without any missing 6-day image pairs. Here we synthetically measure a sinusoidal variation with an amplitude of 50 m/yr, but with gaussian displacement error (standard deviation 2.5 m) added to each synthetic measurement. Despite perfect coverage with 6-day image pairs, the 50 m/yr velocity variation is not clearly evident.

Matlab code to create the plot above is included at the bottom of this response letter.

**...I think the focus of this paper should be optical data, its limitations during polar winters and cloud cover and how your method can still resolve seasonality using optical data.**

The paper is focused almost entirely on optical data, its limitations, and how our method can still resolve seasonality using optical data. However, the method is fully agnostic to whether feature-tracked velocities were obtained from optical or SAR data, so we have been careful to write the paper in a way that does not preclude its application to SAR data.

P30-35. The authors should highlight these significant gaps which still limit our understanding of ice dynamics change on different time scales. I agree that a number of studies on the seasonal ice dynamics of glaciers in Arctic, Antarctic and other glaciated regions are available in bits and pieces, but they do provide a great degree of evidence that help us understand the physical processes which govern the ice dynamics on different time scales. I can't imagine how a consistent global mapping of seasonal ice dynamics looks like, which the authors have pointed to and how, if accomplished, they better our understanding...

The case for consistent, large-scale mapping can be made by drawing a parallel to Rignot et al.'s first comprehensive mapping of Antarctic secular ice velocity, which was published in 2011. Dozens of regional studies of ice velocity had already been published at that time, yet the

application of a consistent measurement technique and synthesis into a single map has already informed nearly a thousand peer reviewed studies. We allude to the potential insights that could be gained from comprehensive mapping with this new sentence, which we have added to the end of the paragraph:

A comprehensive mapping of the world's seasonal ice dynamics would permit direct intercomparison of seasonal evolution in regions with different driving processes; provide a basis for analysis of long-term changes in seasonal behavior; and supply models with a zeroth-order understanding of global ice climatology.

**...It is also not clear why such an approach relies on optical imagery, even though we have a year-around consistent and global SAR imagery by missions like Sentinel-1. These points should be addressed in the Introduction to better form a basis or need for this study.**

The method we present is not limited to optical imagery, and can just as easily be applied to SAR image pairs. Following this suggestion, we have now clarified that point in the abstract by stating,

In this paper, we describe a method of analyzing optical- or SAR-derived feature-tracked velocities...

We reiterate in the final paragraph of the Discussion section,

The methods presented in this paper have focused primarily on optical satellite data because no other type of sensor provides such a long record of ice velocity. As more radar data become available, particularly since the launch of Sentinel 1a/b, the problem of missing winter data will be eliminated, but the methods presented in this paper will still hold.

Figure 7. Nice figure. But when I compared this with Figure 4, I drew a couple of points that need to be addressed. ITS\_LIVE velocity data for Russel Glacier in Greenland is much more dense and appears to be well distributed around the year as compared to Byrd Glacier, Antarctica...

The apparent difference in data density is likely just a matter of figure size. Figure 4 fills the entire page width, and as stated in the caption it shows 14,208 image pairs of Byrd Glacier. In contrast, the panels of Figure 7 are much smaller and show 5189 image pairs of Russell Glacier. We used the same linewidths and marker sizes in both figures, which likely makes the Russell Glacier data appear more dense, given the overall difference in size of the two figures. We include the number of image pairs in the caption to clarify this point.

**...I wonder how such a large number of wintertime velocities are available in Greenland using optical data. Are they averaged for the entire polar wintertime?...**

To be clear, no images have been acquired during any winter at Russell Glacier. This is also true at Byrd Glacier. Here is a normalized histogram of image collection times for both. We've offset the dates of the Russell Glacier images by 183 days to allow direct comparison:

The lower latitude of Russell Glacier (67N) compared to Byrd Glacier (80S) allows a longer image collection season at Russell, but still, months go by each winter without any image acquisitions. If we wish to inspect feature-tracked velocities directly, all we can do is infer average velocities between images captured in the fall and spring, and we display them following the standard convention of using a horizontal line to connect the acquisition times of the two images. We find horizontal lines alone create a confusing mess to inspect visually, so we place a dark dot at the center point of each line. The caption of Figure 4 states,

**Light gray horizontal bars connect the acquisition times of each image...Center dates $t_m$ are shown as dark gray dots for visual clarity.**

In Figures 4 and 7, the horizontal bars that span winters and dark dots placed in their centers may create the impression that data are available during the winter, but that is simply a convention of displaying this type of data because, as far as we know, there isn't a clearer way to visualize it.

...I expect that this dense and well distributed velocity data is the main reason why you have a great sinusoidal fit here, isn't it? Because ideally the method should resolve the missing velocities in winters using the data points for the rest of the year. By the way, it appears to me that ITS\_LIVE observations in case of Russel Glacier are already resolving seasonal variations without applying your proposed method. An example, where velocity data is sparse like for Byrd Glacier, would be much more convincing how well your proposed method can resolve seasonality...

There are only 5189 ITS\_LIVE observations at Russell compared to 14,208 at Byrd. We do not have a years-long record of GPS at Byrd for comparison, but the robustness of our results there makes it clear that the signal is physical, it is persistent, and it can be resolved by our method using any random selection of 1000 or more image pairs.

The central problem our method solves is that velocity variability from feature tracking cannot be interpreted by eye when image pairs are separated by months or more. Figure 1 illustrates this point—the patterns that appear visually in the image pair data in Figures 4 and 7 do not necessarily reflect the underlying patterns of velocity variability, either in amplitude or in phase.

...Russel Glacier is the best case scenario (P270). An example from any mountain glaciers from Alaska, European Alps and high Asia would enhance the applicability of the proposed method. The optical data is available around the year in these regions and is only affected by cloud cover. At present the method is proposed to work in these regions, but the potential challenges are not highlighted.

Russell Glacier is not necessarily a best-case scenario within the global ITS\_LIVE dataset. As far as we can tell, it suffers from rather typical image issues, including summer surface meltwater and cloudiness. It was selected for analysis only because of its known seasonality and the availability of a decade-long GPS record that we could use for validation. The least-squares method we describe does not suffer from winter data gaps, but it does not benefit from the gaps either. As we state in the abstract and in the main text, our method is *agnostic* to data gaps in winter.

**Minor comments:**

P5,25 or elsewhere. Instead of using "accelerations", I would recommend to use "velocity variations" because both acceleration and deceleration are governed by physical processes.

We have replaced the word *accelerations* with the term *velocity variations*.

**P25. Can your approach resolve velocity changes during such short time-scales? If yes, you should highlight in the paper what additional information is required in your method in order to retrieve such signals. If not, there is no need to include it in the Introduction.**

In this introductory paragraph we provide context for the timescales and physical processes that we aim to resolve. We indicate that some previous studies have reported on interannual variability driven by ocean forcing, while other studies have considered the effects of tides on glacier movement. The timescales we wish to resolve are in between fortnightly and interannual, but currently the potential influence of ocean forcing and tides on seasonal timescales are not well understood. We feel it is appropriate to provide this brief background to help readers place this our paper in the context of previous work.

**Figure 1. Example-1 shows a hypothetical scenario, especially the velocity time series shown in blue. We have plenty of SAR data and derived ice velocities for the polar glaciers. What about showing a real case here?**

Real cases are shown in Figures 4 and 7, but visually deciphering what's happening in them is nearly impossible. Rather than overwhelm viewers with thousands of image pairs, we created this simple (but mathematically precise) cartoon to illustrate the fundamental elements of the problem that we solve in this paper.

**P45: satellite image pairs » optical remote sensing image pairs**

The sentence in question reads,

The method we present applies to ice velocity datasets...which have been derived by feature tracking techniques applied to satellite image pairs.

As it is written, the sentence is correct because our method also applies to SAR image pairs.

**P85: It is not clear how the weights are assigned here. Are these based on residuals?**

Line 81-83 states,

We assign the velocity weights  $w_v$  in the polynomial fit using the formal error estimates  $\sigma_v$  from the ITS\_LIVE data such that  $w_v = \sigma_v^{-2}$ .

Figure 4: The colors (blue and green) in the legend and fits are inconsistent.

Fixed.

P105. "Instead, we operate on the displacements associated with each image pair, taken as the integrals of velocities" should be shown as a different figure as this is an important step of the paper. It would be better to see how various displacement estimates at different epochs ranging from days to years are prepared for any sinusoidal fit.

We have taken this suggestion, as we now plot the displacement time series along with the velocity time series to make it a bit more clear. Here is the updated Figure 1 and caption:

The upper panel shows an ice velocity time series in blue, which integrates to form the cumulative displacement time series shown in the lower panel. We use satellite images to measure ice velocity as the cumulative displacement of crevasses and other glacier features that occurs between acquisition times of any two satellite images. Here, four images taken at times  $t_1$  through  $t_4$  provide six unique combinations of image pairs that yield the measurements depicted as horizontal gray lines connecting the acquisition times of each image pair. Vertical gray lines

show measurement uncertainty and a black dot is placed at the center times of each image pair for visual clarity. A dashed sinusoid is fit to velocity measurements at the center times of each image pair to highlight the inadequacy of fitting directly to velocity data for determination of seasonal amplitude or phase. This paper describes an alternate, exact approach, wherein sinusoids are fit to accumulated displacements, which are then converted to velocity to produce the light red velocity sinusoid shown in the upper panel.

**P130. I recommend the authors to make a relationship between Va and Vs here. In other words, the authors should derive equation 4 from above equations or make a relationship between them. What is the goal here? Minimizing the Va?**

We see that we failed to explain how we got to Equation 4. It is just the integral of Equation 1, but that was not made clear. We have modified the text, which now says

If our first estimates of A and  $\varphi$  are correct, then seasonal variability must have aliased each initial estimate of interannual variability by a certain amount  $v_a$  The amount of velocity aliasing equates the seasonal displacement over time, which we can obtain by dividing the definite integral of Eq. 1 by dt, or...

**Matlab code**

% Dates of first image for years of coverage every six days: t = (1:6:365\*2)';

% Dates of second image in each pair: t = [t t+6];

% Displacement error in each image pair (m): sigma\_d = 2.5;

% Corresponding velocity error in each image pair (m/yr): sigma\_v = sigma\_d/(6/365);

% Measurement noise: v\_noise = sigma\_v.\*randn(size(t(:,1)));

% Formal estimates of error (equates to 2.5 m in each image pair): v\_err = sigma\_v.\*ones(size(t(:,1)));

% "Measured" velocity time series with sinusoidal variation of 50 m/yr % amplitude with a max value on day 91 of each year: v = 800+sineval([50 91],t(:,1))+v\_noise;

% Plot the results: itslive\_tsplot(t,v,v\_err) axis tight datetick('x','mmm','keeplimits') box off ylabel 'velocity (m/yr)'

---

## Author Comment (AC2) · 12 Aug 2020

**Reviewer 2**

**General comments**

**The authors present a workflow to fit a sinusoidal function to a data set of clustered velocity estimates on ice sheets and outlet glaciers. The work is well written, and the authors clearly identify the need to extract more concise information of this vast collection of Eath observation data. The steps taken by the authors are explained, but in general there is a tendency to highlight the strong points of the methodology in their argumentation. Being a methodology paper, there might be a reason to keep this presentation brief, but it might be more than worthwhile to emphasis points of improvement and why certain decisions are taken.**

**My main concern with this work stems from the property that the authors define seasonal variation as a cycle. In this way the reader is pushed into a certain narrative, which limits how to approach this issue. The authors are correct about the sinusoidal variation of the forcing (the sun and the seasons), but this does not mean the ice velocity has the same reaction. Personally, I see the seasonal variation more as a perturbation, to which there is a reaction time/response, a peak and fade out/reorganization. Thus a perturbation (including a sinus function, but also a lot of other responses) occurs every year, due to surface melt run-off, but the time span does not need to extend towards a whole year, as is assumed here. If we look at other studies short spikes are clearly visible (Kjeldsen et al. 2017, 10.1002/2017GL074081; Derkacheva et al.2020; 10.3390/rs12121935), or in the dynamics of a surging icecap (Dunse et al. 2015,10.5194/tc-9-197-2015) where a step function is seen, that is initiated by meltwater perturbation. So I miss a discussion on how good a sinus-function is as a model. There is only testing of how good the observations meets the model description, and not how good the model fits the observation. Putting everything on "background interannual variability" is a bit easy**

The primary concern here is that without any regard for the shape, timing, or source of annual forcing mechanisms, we have gone ahead with an assumption that a sinusoid is a reasonable approximation of any seasonal behavior. To be clear, we have not assumed anything about the shape of forcing functions, nor do we directly discuss any potential driving mechanisms in this manuscript. Though we do have understanding of how some types of seasonal velocity signals evolve, currently, our state of knowledge is such that we do not fully understand where seasonal forcing mechanisms exist, what they are, or what their shapes may be.

We do, however, provide a method for gaining insights into a glacier's response to seasonal variability in forcing, and for this we use a sinusoid. The sinusoid does not assume anything about the shape of the forcing mechanism or even the shape of glacier's response. Rather, the sinusoid *describes* the cyclic behavior in the simplest way possible.

We contend that we must understand the most basic level of behavior before we can begin to discuss aberrations from it. This means that before we can begin fitting higher order functions or investigating how seasonal cycles change from year to year, we must first identify *where*

seasonal variability exists, *how significant* it is in terms of the overall displacement cycle of the glacier, and in *what season* of the year ice velocity tends to reach its maximum.

The value of the sinusoidal approach can be seen in a new preprint by Riel et al. (https://doi.org/10.5194/tc-2020-193), which was submitted to *The Cryosphere* after the reviews for our manuscript were posted. Using the exact approach we describe, they map the seasonal amplitude and phase of variability to observe traveling waves in Sermeq Kujalleq. The map here provides evidence for kinematic behavior that begins at the glacier terminus and travels upstream each year.

[Figure]

Sermeq Kujalleq does not exhibit perfectly sinusoidal behavior, but the simple two-term description of seasonal variability reduces complexity and makes interpretation straightforward. If more complex terms are desired, we point out in the paper that additional sinusoids can be added, and it eventually it will be possible to build a Fourier series with this approach.

**Another question arising is the wording of climatological velocity, I am not able to figure out what the authors mean with this term. This directly also brings me to a second point on the sinus fitting, as it is treated as a cyclic function similar to (Menchew etal., 2017, 10.1002/2016JF003971). They look at a tidal time span, where the forces are highly repetitive in magnitude and phase. However, if this is the case for seasonal glacier velocities is not so clear, as the amplitude of glacier velocity seem to correlate with surface mass balance. This has been observed with GPS in Greenland (van deWal et al. 2015, 10.5194/tc-9-603-2015) or on Nordenskiöldbreen, Svalbard (van Peltet al. 2018 10.1029/2018GL077252). But the sinus function of the authors does not take the change in amplitude, from year to year into account. However, this (to me) would be a climatological velocity (if I had to guess what the authors mean).**

We have clarified the definition of climatology with the addition of this sentence:

*In this paper, we describe a robust method of measuring the* climatology—*or average seasonal cycle—of ice flow dynamics, with the ultimate goal that our method may be used to map the typical magnitude and timing of the seasonal glacier dynamics worldwide.*

**Other influencing phenomena, like the ocean/front position have similar seasonal amplitude change (Kehrl et al. 2016, 10.1002/2016JF004133). By putting all these into a cyclic function, the signals of phase and amplitude might smooth out. In connection to this, at high latitudes, the coverage is concentrated towards the summer season. Hence, how do short term perturbation propagate into the velocity estimation? From the synthetic test the methodology can be "considered agnostic", but this is true for reconstruction purposes of a sinusoidal function. It is also not clear where the authors are after, the onset of speed-up, or "identify the seasonal maximum velocity"? Other studies/data/methods are able to find the timing of such speed-up events (Altena et al.2017 10.3389/feart.2017.00053, Vijay et al. 2019, 10.1029/2018GL081503), though not as precise or automatic as presented here, but are less constrained. So, there are some issues on the amplitude, but also on the phase. The argument of the authors for using a sinus, as it is "elegant" is a bit weak in my opinion. It would very much strengthen the manuscript, if these influencing effects/considerations are highlighted, as it gives handholds on the way forward.**

We clearly hear the reviewers concern that a sinusoid is not a perfect representation of glacier variability and that in some cases it may even be a poor representation. We completely agree with this assessment but disagree that a sinusoidal approximation is a bad first guess in the absence of a universal model of variability. We could assume a sawtooth or step function, or even a pricewise model but we're unconvinced that any of these models would not suffer from the same shortcomings. The problem is how can we use heterogeneous data to compare across vastly different climate conditions, glacier characteristics, ocean forcing, etc., to identify where glaciers are fluctuating seasonally. A sinusoid is the most basic assumption we can make and provides a starting point for the contextualization of global glacier variability that will increase in nuance and complexity with time. We justify the simplicity of our assumption as it is highly generic and provides a logical first step toward global characterization.

**The authors have formulated their estimation procedure by decoupling the x- and y-velocities. Is there a certain reason for this? I can imagine it can be beneficial, to include co-variance functions, so outliers in one dimension are also excluded in the other. In addition, given these phase angles are estimated independently, do the authors see a difference between both axis. If so, this would imply a change inflow direction, if not what would that mean? Also, why did the authors do filtering (using the MAD), and not do robust least squares, or at least use such procedure in the estimation? Neither is it clear to me why several iterations are applied, see (https://ccrma.stanford.edu/~jos/filters/Sum_Sinusoids_Same_Frequency.html), or is the estimation not restricted to a yearly cycle? Or is the iteration not done on the residuals?**

Some of the most compelling insights that we expect to gain from this method will involve transverse motion that could not be detected if we were to assume that flow variations only occur

in the direction of mean flow. For example, we have applied our method to Drygalski Ice Tongue and found the same side-to-side motion that has previously been found using in-situ measurements (https://ui.adsabs.harvard.edu/abs/2013AGUFM.C21A0624L/abstract).

In addition to floating ice, we may find transverse flow anomalies in grounded areas with strong seasonality of basal water pressure. For example, if the basal pressure on one side of a glacier rises while the other side remains unchanged, the lopsided acceleration in flow would cause a divergence of and change in flow direction, even if only small. The method we describe is able to resolve displacements of just a few meters left or right of a mean flowline, meaning the maps can be created by separating the $x$ and $y$ components of variability could hint at underlying drivers of change.

**The authors run tests on synthetic data, by imposing corruption to individual velocity estimates. This noise is done on an individual basis, which is partly due to measurement noise. But there is also dependent noise, as displacement estimates are derived for pairs of images. Hence, when one image is corrupted for some reason, there is a high probability it propagates to all displacements it is part of. However, this issue is not included into the analysis, though of importance (and due to the synthetic nature, is possible to generate). This would give more insights then the 32 velocity estimates, stated now.**

The reviewer makes a very good point but there is a subtlety to the data that makes this not the case. The largest source of error when tracking features between two images acquired with the same viewing geometry (repeat image) is the geolocation error in each image that typically manifests itself as a scalar displacement in $x$ and $y$. These correlated errors have been corrected by the autoRIFT algorithm that produces the ITS_LIVE velocity data. The correction process works by examining the initial measured velocities over all stable surfaces in an image such as rock. The average measured velocity over rock is equates to an offset error across the entire scene. After the offset errors are removed in $x$ and $y$, the remaining errors in each image pair can be considered to be uncorrelated.

**Minor comments:**

**In general the manuscript is well written, the authors write in their mother tongue, so concerning this issue I am not able to do any better. But for a global audience the wording is sometimes a bit hard; I have learned quite a lot of new words. For sake of easy reading, and not having to go back and forth to a dictionary, please consider changing words a bit. Think of, "unwieldy" or "egregious".**

We appreciate this feedback, as we wish to make our work accessible to all interested readers, regardless of personal background. We have looked through the manuscript to ensure that (aside from a few necessary technical words) there is no language in this revision that wouldn't be found in in standard English-language news outlets.

**I have tried to understand from the text what is done, and also looked in the code to be able to zoom into the plots/data. But the provided code and plotting does not work, as some functions ("itslive_tsplot" or "itslive_seasonal_deets") are absent.**

We appreciate the reviewer's effort in digging into the code that we included as a supplement to the manuscript. It appears this comment regards the make_fig04.m script, which contains the code that can be used to recreate Figure 4 of the manuscript. We've double-checked and cannot identify any missing functions, but it seems likely that the confusion stems from our inclusion of the itslive_seasonal_deets function at the end of the make_fig04.m script. It's a relatively new feature that Matlab can call functions that are included at the end of a standard non-function .m script, and we've taken advantage of the feature to keep our bundle of supplemental code as tidy as possible.

Regarding the missing itslive_tsplot function, we note that it is included among the ITS_LIVE functions we've posted to GitHub (www.github.com/chadagreene/ITS_LIVE). The README.txt file that describes what's included in the supplemental material states at the top that some functions necessary to run the scripts are part of the toolboxes on my GitHub page, including ITS_LIVE tools, Antarctic Mapping Tools for Matlab (Greene et al., 2017), and the Climate Data Toolbox for Matlab (Greene et al., 2019).

**title: be a bit more specific, maybe change to "Reconstructing seasonal oscillations" also include "glacier ice".**

We note that the application of this method is not limited to glaciers. For example, we have applied our method to the Ross Ice Shelf and found the same patterns of seasonal variability reported this week by Klein et al. (https://doi.org/10.1017/jog.2020.6).

**6: "dark polar winters" > "at high latitudes"**

The sentence in question describes the problem of observing seasonal variability using optical data, when no optical data are available for several months each year due to lack of sunlight. Thus, we describe the situation that there are "...no optical observations throughout *dark, polar winters*." In our view, the suggested wording, "*at high latitudes*" does not directly address seasonality nor make mention of the solar illumination that's necessary for optical data acquisition. We prefer to keep the wording as it is.

**8: "climatological average winter velocities" what is meant here?**

We have clarified the definition of climatology with the addition of this sentence:

*In this paper, we describe a robust method of measuring the* climatology—*or average seasonal cycle—of ice flow dynamics, with the ultimate goal that our method may be used to map the typical magnitude and timing of the seasonal glacier dynamics worldwide.*

**15: "sufficient quantity of data" is this due to quantity of data, the consistency of campaigns/ monitoring programs or simple availability of large computing power. Or is it opening up of the archives, making historical flow estimation possible (Cheng et al.2019 10.5194/isprs-archives-XLII-2-W13-1735-2019).**

The conference paper by Cheng et al is intriguing, and if they are able to successfully employ the method of processing outlined in that paper, it will be interesting to see what the velocity fields from the ARGON era look like. But as far as we can tell, that dataset has not been produced or has not been made widely available.

**18: "all of the world's ice" large bodies of glacial ice**

We have clarified that annual velocity mosaics are now available for "*nearly* all of the world's *land* ice".

**20: "upended glaciology" Remote sensing is able to get geometric information about the (sub)surface, and is of great aid. To some extent this is a game changer, but it might be fair to also give credit to automatic weather stations, or put it into persective. This have been other great advancement in glaciology, think for example of (Ohmuraet al., 1992; Oerlemans et al., 1998).**

The introductory paragraph briefly mentions some of the recent advancements in remote sensing that have gotten us where we are today, and then the paragraph identifies the types of insights we want to gain from this new abundance of satellite data. We feel that a discussion of automatic weather stations would be a distraction from the main points we wish to communicate in this manuscript.

**31: There is also an large collection of studies at the intermediate timescale, which isleft out here, dealing with surges. For sake of completeness, this might be included**

We do reference a paper on surge dynamics by Yasuda and Furuya, but we have tried to keep the focus of this manuscript primarily on cyclic behavior.

**37: "the logistical challenge of" what is meant here?**

We have modified the sentence to now read,

*...due in part to the **technical** challenge of working with optical data in polar regions, where the surface is not touched by sunlight for months-long periods each winter.*

The technical challenge of working with optical data in polar regions is that several months go by each winter when optical data are not collected, because the sun does not illuminate the surface during those months. The full text of this manuscript provides a detailed description of the technical challenges of working with this data.

**39: "robust extraction" using a robust pre-processing technique, is different from a robust methodology. Given its stiffness (non adaptive) towards one model (a sinusoidal) this might not be a correct formulation. It might be "precise"...?**

Following this suggestion, we have changed the word *robust* to *precise*.

**40: "primarily focused on Antarctica" maybe better: on the ice sheets and their outlets....?**

We designed the study with the primary goal of understanding Antarctic seasonal ice dynamics, because that's where data is most limited, and it's where the fewest studies have been published about the subject. The examples we provide and the statistics in all of the synthetic tests are based on Antarctic data. Accordingly, we stand by the statement that

*Our study is primarily focused on Antarctica, where seasonal variability is poorly understood, and where data limitations currently present the greatest challenges to making such measurements.*

**53: "by feature tracking" add: over longer time spans**

We have made the suggested change.

**55: "the true magnitude" change to "a" instead of "true" or "a well fit"**

We have removed the phrase true magnitude, as suggested. The sentence now reads,

*...by fitting a cyclic function to the time series of displacements rather than average velocities, we show that it is possible to accurately recover the magnitude and phase of seasonal velocity variability.*

**56: I miss another possibility here, which is common practice in inSAR displacement estimation, being inversion (e.g.: Bontemps et al. 2018, 10.1016/j.rse.2018.02.023, Li et al. 2020, 10.1016/j.rse.2020.111695). This does not make it necessary to work with average time stamps or a model, and can resolves to very small time steps.**

It is unclear how the inversion techniques described by Bontemps et al. or Li et al. should be included in this study.

**64: " first or second image" first and second? maybe be more clear or use "master-slave" "chip-search space" etc.**

We note that the community is moving away from the terms *master* and *slave* (https://comet.nerc.ac.uk/about-comet/insar-terminology/).

The sentence in question states,

*Each satellite image may serve as the first or second image in multiple image pairs...*

The words "first" and "second" refer to the sequence in time. The suggested wording does not adequately convey temporal sequence, and the meaning of "chip-search space" may not be widely understood.

**100: "most robust means" why is this robust, where do you get the reliability from?**

We have removed the word "robust".

**fig4: maybe it is good to note, why there are two groups of points, as one is +- a year and the other at short time intervals in summer. btw: the purple line nicely follows the annual velocity clusters!**

Following this suggestion, we have edited the caption of Figure 4 to describe the two groups of points as follows:

*The clustering of these 14,208 measurements taken near the grounding line of Byrd Glacier typifies ITS_LIVE image pair data, with short Δt measurements providing direct, but noisy observations of velocity variability throughout each summer, while much lower-noise winter estimates can only give insight into the total displacement that occurs during the dark, winter months.*

**151: I don't think this is sensitivity, but more an analysis to get an idea how good the recovery is. As the model is corrupted with noise and then an attempt is made to reconstruct the model. If I understand correctly.**

The Sensitivity Analysis section is where we test the sensitivity of the method to several different parameters. We determine the sensitivity of the technique to the number of image pairs used, the level of background interannual variability, the amplitude of the underlying signal, and the phase of the underlying signal. The parameters of these tests are tabulated in Table 1: Sensitivity test parameters.

**181: "recover", change to we "are able to describe the variation by seasonal cycles" or alike.**

The passage in question reads,

*We conducted several tests to determine the accuracy with which we can recover the amplitudes and phases of seasonal cycles in synthetic velocity time series.*

We feel that the present wording will be more easily understood than the suggested wording.

**258: " is remarkable" subjective wording, please change**

Following this suggestion, we have changed the word *remarkable* to *notable*.

**258: "robust", precise/accurate might fit better**

Following this suggestion, we have changed the word *robust* to *precise*.

**267: "minuscule variations" subjective wording**

We have replaced the word *minuscule* with *tiny*, to describe the displacements on the order of a meter or so that can be detected with this method.

**283: " in the climatological sense, nature does not consistently time such events as calving or increases in basal water pressure with any greater precision than the method we have presented to detect them". What is meant here?**

We have added a definition of climatology to clarify that the climatology refers to the average seasonal cycle taken over many years.

Transient events such as calving or impulses of water into a subglacial hydrological system often occur on a yearly cycle, but the corresponding glacier speedup may only last for a few days. It may seem that a spike in velocity only lasting a few days of the year would be poorly represented by a sinusoid that continuously varies throughout the year, but this sentence points out that mother nature does not time glacier calving to occur on the very same day each year. As a result, when we take the average annual cycle from many years of data, we find there is generally a *season* of glacier acceleration rather than just a few days of acceleration.

To illustrate this point, here we consider a glacier whose velocity is exactly 800 m/yr most days of the year, but each summer it accelerates by 50±10 m/yr for a duration of 10±3 days, centered on June 1±20 days. We generate 1000 years of this pattern, and then consider the mean cycle of daily speeds and compare it to the sinsuoid fit. Using the Matlab code provided at the bottom of this document, we get the following plot:

[Figure]

What we see in the daily mean velocity curve is that in the climatological sense, the period of high summer velocity lasts for months, even though the average high-velocity event only lasts for 10 days. Fitting a sinusoid to the daily means finds a peak velocity on June 1, which is exactly the prescribed center date of the high-velocity period. Of course, the amplitude is severely underestimated by the sinusoid in this scenario, but the passage in question regards timing, not amplitude. It states,

*While it is true that a glacier can accelerate in response to a transient event and return to an equilibrium velocity within just a few days (Stevens et al., 2015; Andrews et al., 2014), in the climatological sense, nature does not consistently time such events as calving or increases in basal water pressure with any greater precision than the method we have presented to detect them.*

**285: "In most cases, a sinusoid will likely capture the majority of velocity variance throughout the year, and represent the fundamental mode of subannual variability in ice velocity." Please justify this claim, as this is the corner stone which the whole study is build upon.**

We agree that this claim was not well supported, and on reflection we see that it may have been untrue. We have removed the claim from the text.

**291+: It seems the authors put all the misfits of the sinus model on the inaccuracies of the GPS measurements, while this sensor measures all kinds of physics decadal, annual, daily,...**

We do not claim that the small misfit between GPS and our method is solely the fault of the GPS data. Rather, we simply point out that the GPS record contained several long gaps, and it is possible that any disagreement between the two methods could reflect the different times during which data were collected. Here in the Discussion section it is appropriate to acknowledge the possible causes of any mismatch between the GPS and ITS_LIVE data, and consider what that might mean for applying this method elsewhere. The passage states,

*...the [GPS] receiver's harsh polar environment has led to several long gaps during which no GPS position data were acquired. This suggests that as a means of measuring ice dynamic climatology, our method might not only meet, but exceed the performance of the in situ GPS receiver while providing insights into dynamic behavior as far back as the mid 1980s.*

We feel it is important to point out that GPS data—although absolutely vital to this type of work—is not perfect. Most GPS receivers do not collect data over polar winter, yet the method of image analysis we describe is able to approximate winter ice velocities. Similarly, very few locations on Earth offer GPS measurements even today, yet the ITS_LIVE dataset offers global coverage, and in some locations that coverage extends back to the 1980s.

**305: "our method can extract" please add "..by describing ice flow as an oscillation ..."in some way**

Following the suggestion, the passage now reads,

*...our method can extract the amplitude of seasonal variability with a precision on the order of about 1 m/yr **by describing ice flow as an oscillation**, provided the level of background interannual variability does not overwhelm the overall signal.*

**309: "independent of the amplitude and phase of the seasonal velocity variability", this is not convincingly given.**

The sentence in question reads,

*Ability to detect seasonal amplitudes is independent of the amplitude and phase of the seasonal velocity variability, but phase accuracy benefits with increasing amplitude of seasonal variability.*

Please see Figures 5g, 5i, and 6a.

**313: "fully three dimensional understanding" what is meant here?**

We have fixed this sentence. It now reads,

*...by providing a method that can be employed independently in the two dimensions of Cartesian coordinates, we hope to gain a **more complete** understanding of how dynamic signals propagate through the world's ice.*

**322: "egregious outliers" egregious=outliers**

We have replaced *egregious* with *extreme* to make the text more accessible to non-native English speakers. To be clear, we are not discussing outliers that are just three or four standard deviations away from the mean. Rather, we are discussing the extreme outliers that can land hundreds of standard deviations away from the rest of the bunch. We now describe them as *extreme* outliers.

**Matlab code**

```matlab
% Create synthetic time series:
N = 1000; % number of years of the time series
dur = 10 + round(3*randn(N,1)); % (days) duration of fast flow each summer
dur(dur<1) = 1; % just in case randn resulted in any negative durations.
st = 153 + round(20*randn(N,1)-dur/2); % DOY of fast flow period.
vs = 50+20*randn(N,1); % Magnitude of summer speedup
vs(vs<5) = 5;  % just in case randn made vs negative
va = 800; % background/winter ice velocity

% Define the rectangular function for each year:
v = va*ones(N,365);
for k = 1:N
   ind = st(k):(st(k)+dur(k)); % indices corresponding to fast-flow event
   v(k,ind) = va+vs(k);
end

%% Plot the time series

cm = rand(N,3); % colormap
figure
hold on
for k = 1:N
   hi(k) = plot(1:365,v(k,:),'color',cm(k,:));
end
axis tight
box off
hm = plot(1:365,mean(v),'k','linewidth',2);

% Fit a sinusoid:
ft = sinefit(1:365,mean(v),'terms',3) % Climate Data Tools (Greene et al.,
2019)
hf = plot(1:365,sineval(ft,1:365),'r','linewidth',2);

datetick('x','mmm','keeplimits')
ylabel 'glacier speed (m/yr)'

legend([hi(1),hm,hf],'annual data','daily mean','sinusoid
fit','location','northwest')
legend boxoff
```

---

## Referee Report (RR1)

My response has a strong delay, as the second review underwent several iterations. However, I have now a better understanding of the intentions of the authors. It will benefit the work if the authors are more clear about their objective. To my understanding they are expanding upon the work of Armstrong et al. 2018, which looked at speed-ups and slow-downs on a regional scale. The authors are merging multiple years of data, to get a general or overall timing and magnitude of this variations. Appearantly, the authors have formulated this in their discussion, but repetition in earlier parts of the work will guide the reader.

In this respect I think the wording of climatological velocity is misleading, as it is pushing the reader towards other associations. Firtstly, because it gives an appearant link to the phenomena, while this cause can also be specific or attributed to the local configuration. Secondly, it introduces a term which can be confused with the climatological mass balance, see the glossary of IACS.

Furthermore, the response of the authors is sometimes a bit frustrating, as on multiple occassions the authors repond to the first or last part of a question. Neglecting the other points mentioned. For only a handfull of instances the authors refer to other work to strengthen their argument.

In the minor comments the authors have the tendency to keep the text as is. Generally, they take the argument to make it easier to understand and read for the general audience, as the readers of The Cryosphere are ill-informed...

**RC1:** The authors present a workflow to fit a sinusoidal function to a data set of clustered velocity estimates on ice sheets and outlet glaciers. The work is well written, and the authors clearly identify the need to extract more concise information of this vast collection of Eath observation data. The steps taken by the authors are explained, but in general there is a tendency to highlight the strong points of the methodology in their argumentation. Being a methodology paper, there might be a reason to keep this presentation brief, but it might be more than worthwhile to emphasis points of improvement and why certain decisions are taken.

My main concern with this work stems from the property that the authors define seasonal variation as a cycle. In this way the reader is pushed into a certain narrative, which limits how to approach this issue. The authors are correct about the sinusoidal variation of the forcing (the sun and the seasons), but this does not mean the ice velocity has the same reaction. Personally, I see the seasonal variation more as a perturbation, to which there is a reaction time/response, a peak and fade out/reorganization. Thus a perturbation (including

a sinus function, but also a lot of other responses) occurs every year, due to surface melt run-off, but the time span does not need to extend towards a whole year, as is assumed here. If we look at other studies short spikes are clearly visible (Kjeldsen et al. 2017, 10.1002/2017GL074081; Derkacheva et al.2020; 10.3390/rs12121935), or in the dynamics of a surging icecap (Dunse et al. 2015,10.5194/tc-9-197-2015) where a step function is seen, that is initiated by meltwater perturbation. So I miss a discussion on how good a sinus- function is as a model. There is only testing of how good the observations meets the model description, and not how good the model fits the observation. Putting everything on "background interannual variability" is a bit easy.

**AC1:** The primary concern here is that without any regard for the shape, timing, or source of annual forcing mechanisms, we have gone ahead with an assumption that a sinusoid is a reasonable approximation of any seasonal behavior. To be clear, we have not assumed anything about the shape of forcing functions, nor do we directly discuss any potential driving mechanisms in this manuscript. Though we do have understanding of how some types of seasonal velocity signals evolve, currently, our state of knowledge is such that we do not fully understand where seasonal forcing mechanisms exist, what they are, or what their shapes may be.

We do, however, provide a method for gaining insights into a glaciers response to seasonal variability in forcing, and for this we use a sinusoid. The sinusoid does not assume anything about the shape of the forcing mechanism or even the shape of glaciers response. Rather, the sinusoid *describes* the cyclic behavior in the simplest way possible.

We contend that we must understand the most basic level of behavior before we can begin to discuss aberrations from it. This means that before we can begin fitting higher order functions or investigating how seasonal cycles change from year to year, we must first identify where seasonal variability exists, how significant it is in terms of the overall displacement cycle of the glacier, and in what season of the year ice velocity tends to reach its maximum.

**RC2:** You can pick on my formulation, but this deviates from the point raised here. There is no test if the cyclic behavior of your function *describing* the glacier response is correct. If there is a biased sampling in a speed-up or slow-down period, the seassonal estimates of the maximum flow period might be off.

There are statistical tests if your model describes the observations properly. Why are such describtors not assessed here, such as a Generalized Likelihood Test, or other test

statistics?

An annual sinoisdal function imposes constrains on the glacier response: the response time needs to be exactly one year. While this does not need to be the case.

**AC1:** The value of the sinusoidal approach can be seen in a new preprint by Riel et al. (https://doi.org/10.5194/tc-2020-193), which was submitted to The Cryosphere after the reviews for our manuscript were posted. Using the exact approach we describe, they map the seasonal amplitude and phase of variability to observe traveling waves in Sermeq Kujalleq. The map here provides evidence for kinematic behavior that begins at the glacier terminus and travels upstream each year.

Sermeq Kujalleq does not exhibit perfectly sinusoidal behavior, but the simple two-term description of seasonal variability reduces complexity and makes interpretation straight-forward. If more complex terms are desired, we point out in the paper that additional sinusoids can be added, and it eventually it will be possible to build a Fourier series with this approach.

**RC2:** The study of Riel et al. is by no means using sinsoidal functions but b-splines. This is a function that has less constrains to the describtion of its flow. Why do the authors put a single focus on Fourier series, and strengthen their argument by showing an implementation of an adaptive and data-driven function.

**RC1:** Another question arising is the wording of climatological velocity, I am not able to figure out what the authors mean with this term. This directly also brings me to a second point on the sinus fitting, as it is treated as a cyclic function similar to (Menchew etal., 2017, 10.1002/2016JF003971). They look at a tidal time span, where the forces are highly repetitive in magnitude and phase. However, if this is the case for seasonal glacier velocities is not so clear, as the amplitude of glacier velocity seem to correlate with surface mass balance. This has been observed with GPS in Greenland (van deWal et al. 2015, 10.5194/tc- 9-603-2015) or on Nordenskildbreen, Svalbard (van Peltet al. 2018 10.1029/2018GL077252). But the sinus function of the authors does not take the change in amplitude, from year to year into account. However, this (to me) would be a climatological velocity (if I had to guess what the authors mean).

**AC1:** We have clarified the definition of climatology with the addition of this sentence: *In this paper, we describe a robust method of measuring the climatologyor average seasonal cycleof ice flow dynamics, with the ultimate goal that our method may be used to*

*map the typical magnitude and timing of the seasonal glacier dynamics worldwide.*

**RC2:** Climate is based on measurements of least 30 years, how does annual sinusoidal fitting come to this. Please remove this term, as this is now confusing. The additional words do bring some clarification, but is minimal. Only inserting the word "typical" might not be sufficient.

Why do the authors use robust, when no reliability parameters are used. While fitting the describing function the procedure might be stable against outliers, but this does not make it robust.

The suggested work can be a starting point to describe why you deviate from such studies. What the intended objective is, and why inter annual deviations from the sinusoid are not the intended objective (i.e.: not reconstruction, but first order description).

**RC1:** Other influencing phenomena, like the ocean/front position have similar seasonal amplitude change (Kehrl et al. 2016, 10.1002/2016JF004133). By putting all these into a cyclic function, the signals of phase and amplitude might smooth out. In connection to this, at high latitudes, the coverage is concentrated towards the summer season. Hence, how do short term perturbation propagate into the velocity estimation? From the synthetic test the methodology can be "considered agnostic", but this is true for reconstruction purposes of a sinusoidal function. It is also not clear where the authors are after, the onset of speed-up, or "identify the seasonal maximum velocity"? Other studies/data/methods are able to find the timing of such speed-up events (Altena et al.2017 10.3389/feart.2017.00053, Vijay et al. 2019, 10.1029/2018GL081503), though not as precise or automatic as presented here, but are less constrained. So, there are some issues on the amplitude, but also on the phase. The argument of the authors for using a sinus, as it is "elegant" is a bit weak in my opinion. It would very much strengthen the manuscript, if these influencing effects/considerations are highlighted, as it gives handholds on the way forward.

**AC1:** We clearly hear the reviewers concern that a sinusoid is not a perfect representation of glacier variability and that in some cases it may even be a poor representation. We completely agree with this assessment but disagree that a sinusoidal approximation is a bad first guess in the absence of a universal model of variability. We could assume a sawtooth or step function, or even a pricewise model but were unconvinced that any of these models would not suffer from the same shortcomings. The problem is how can we use heterogeneous data to compare across vastly different climate conditions, glacier

characteristics, ocean forcing, etc., to identify where glaciers are fluctuating seasonally. A sinusoid is the most basic assumption we can make and provides a starting point for the contextualization of global glacier variability that will increase in nuance and complexity with time. We justify the simplicity of our assumption as it is highly generic and provides a logical first step toward global characterization.

If this concern is agreed upn, why is this then not clearly reflected in the manuscript. Some improvements in the discussion section might help.

**RC1:** The authors have formulated their estimation procedure by decoupling the x- and y- velocities. Is there a certain reason for this? I can imagine it can be beneficial, to include co-variance functions, so outliers in one dimension are also excluded in the other. In addition, given these phase angles are estimated independently, do the authors see a difference between both axis. If so, this would imply a change inflow direction, if not what would that mean?

**AC1:** Some of the most compelling insights that we expect to gain from this method will involve transverse motion that could not be detected if we were to assume that flow variations only occur in the direction of mean flow. For example, we have applied our method to Drygalski Ice Tongue and found the same side-to-side motion that has previously been found using in-situ measurements (`https://ui.adsabs.harvard.edu/abs/2013AGUFM.C21A0624L/abstract`).

In addition to floating ice, we may find transverse flow anomalies in grounded areas with strong seasonality of basal water pressure. For example, if the basal pressure on one side of a glacier rises while the other side remains unchanged, the lopsided acceleration in flow would cause a divergence of and change in flow direction, even if only small. The method we describe is able to resolve displacements of just a few meters left or right of a mean flowline, meaning the maps can be created by separating the x and y components of variability could hint at underlying drivers of change.

**RC2:** Very nice, are such considerations now also included in the discussion? This helps understanding why such decoupling is done. Nonetheless, making the claim your methods "builds a two dimensional understanding of how ice moves throughout the year", is maybe optimistic (p6 l103).

**RC1:** Also, why did the authors do filtering (using the MAD), and not do robust least squares, or at least use such procedure in the estimation? Neither is it clear to me why several iterations are applied, see (`https://ccrma.stanford.edu/jos/filters/Sum_Sinusoids_Same_Frequency.html`), or is the estimation not restricted to a yearly cycle? Or is the iteration not done on the residuals?

**RC2:** could you please answer these questions?

**RC2:** Is this because of the outliers. The authors use thousands of velocity fields to fit their sinusoid. Are so many velocity fields needed? How much is dependent on the outlier rate? When does this not work anymore, at 10 precent, 30 precent?

**RC1:** The authors run tests on synthetic data, by imposing corruption to individual velocity estimates. This noise is done on an individual basis, which is partly due to measurement noise. But there is also dependent noise, as displacement estimates are derived for pairs of images. Hence, when one image is corrupted for some reason, there is a high probability it propagates to all displacements it is part of. However, this issue is not included into the analysis, though of importance (and due to the synthetic nature, is possible to generate). This would give more insights then the 32 velocity estimates, stated now.

**AC1:** The reviewer makes a very good point but there is a subtlety to the data that makes this not the case. The largest source of error when tracking features between two images acquired with the same viewing geometry (repeat image) is the geolocation error in each image that typically manifests itself as a scalar displacement in x and y. These correlated errors have been corrected by the autoRIFT algorithm that produces the ITS_LIVE velocity data. The correction process works by examining the initial measured velocities over all stable surfaces in an image such as rock. The average measured velocity over rock is equates to an offset error across the entire scene. After the offset errors are removed in x and y, the remaining errors in each image pair can be considered to be uncorrelated.

**RC1:** The authors use optical imagery and thus rely upon the appearance of the surface to infer ice velocity. However an image might be corrupted, causing not white noise, but an offset. To the best of my knowledge, this does happen, and by no means is a subtlety. If looked at the covariance matrix, there are therfore off-diagonal entities, which implies systematic errors. I hope the authors understand what is meant here.

**Minor comments:**
**RC1:** In general the manuscript is well written, the authors write in their mother tongue, so concerning this issue I am not able to do any better. But for a global audience the wording is sometimes a bit hard; I have learned quite a lot of new words. For sake of easy reading,

and not having to go back and forth to a dictionary, please consider changing words a bit. Think of, "unwieldy" or "egregious".

**AC1:** We appreciate this feedback, as we wish to make our work accessible to all interested readers, regardless of personal background. We have looked through the manuscript to ensure that (aside from a few necessary technical words) there is no language in this revision that wouldnt be found in in standard English-language news outlets.

**RC1:** I have tried to understand from the text what is done, and also looked in the code to be able to zoom into the plots/data. But the provided code and plotting does not work, as some functions ("itslive_tsplot" or "itslive_seasonal_deets") are absent."

**AC1:** We appreciate the reviewers effort in digging into the code that we included as a supplement to the manuscript. It appears this comment regards the make_fig04.m script, which contains the code that can be used to recreate Figure 4 of the manuscript. Weve double-checked and cannot identify any missing functions, but it seems likely that the confusion stems from our inclusion of the itslive_seasonal_deets function at the end of the make_fig04.m script. Its a relatively new feature that Matlab can call functions that are included at the end of a standard non-function .m script, and weve taken advantage of the feature to keep our bundle of supplemental code as tidy as possible.

Regarding the missing itslive_tsplot function, we note that it is included among the ITS_LIVE functions weve posted to GitHub (www.github.com/chadagreene/ITS_LIVE). The README.txt file that describes whats included in the supplemental material states at the top that some functions necessary to run the scripts are part of the toolboxes on my GitHub page, including ITS_LIVE tools, Antarctic Mapping Tools for Matlab (Greene et al., 2017), and the Climate Data Toolbox for Matlab (Greene et al., 2019).

**RC1:** title: be a bit more specific, maybe change to "Reconstructing seasonal oscillations"

**RC2:** Has a clearer title been found?

**RC1:** also include "glacier ice".

**AC1:** We note that the application of this method is not limited to glaciers. For example, we have applied our method to the Ross Ice Shelf and found the same patterns of seasonal variability reported this week by Klein et al. (https://doi.org/10.1017/jog.2020.6).

**RC2:** surely the methodology can be applied to ice shelves, but the term "glacier ice" is more suggested to distinguish from sea-ice, which is a topic of many readers of The Cryosphere as well.

**RC1:** 6: "dark polar winters" → "at high latitudes"

**AC1:** The sentence in question describes the problem of observing seasonal variability using optical data, when no optical data are available for several months each year due to lack of sunlight. Thus, we describe the situation that there are "...no optical observations throughout dark, polar winters." In our view, the suggested wording, "at high latitudes" does not directly address seasonality nor make mention of the solar illumination thats necessary for optical data acquisition. We prefer to keep the wording as it is.

**RC2:** Sure, but the confusing is caused because the authors sometimes focus on Antarctica. While at other instances talk about all the ice in the world.

**RC1:** 8: "climatological average winter velocities" what is meant here?

**AC1:** We have clarified the definition of climatology with the addition of this sentence: In this paper, we describe a robust method of measuring the climatologyor average seasonal cycleof ice flow dynamics, with the ultimate goal that our method may be used to map the typical magnitude and timing of the seasonal glacier dynamics worldwide.

**RC2:** This is not sufficient, what is meant here. Otherwise remove this formulation/term.

**RC1:** 15: "sufficient quantity of data" is this due to quantity of data, the consistency of campaigns/ monitoring programs or simple availability of large computing power.

"Sufficient" is a nice word, but does not give much insights here. While the introduction has a whole descibtion on the history of velocity extraction, here the authors only put sufficient in. But how come this is the case then? Otherwise remove.

**RC1:** 15: Or is it opening up of the archives, making historical flow estimation possible (Cheng et al.2019 10.5194/isprs-archives-XLII-2-W13-1735-2019).

**AC1:** The conference paper by Cheng et al is intriguing, and if they are able to successfully employ the method of processing outlined in that paper, it will be interesting to see what the velocity fields from the ARGON era look like. But as far as we can tell, that dataset has not been produced or has not been made widely available.

**RC1:** 18: "all of the worlds ice" large bodies of glacial ice

**AC1:** We have clarified that annual velocity mosaics are now available for "nearly all of the worlds land ice".

**RC1:** 20: "upended glaciology" Remote sensing is able to get geometric information about the (sub)surface, and is of great aid. To some extent this is a game changer, but it might be fair to also give credit to automatic weather stations, or put it into perscetive. This have been other great advancement in glaciology, think for example of (Ohmuraet al., 1992; Oerlemans et al., 1998).

**AC1:** The introductory paragraph briefly mentions some of the recent advancements in remote sensing that have gotten us where we are today, and then the paragraph identifies the types of insights we want to gain from this new abundance of satellite data. We feel that a discussion of automatic weather stations would be a distraction from the main points we wish to communicate in this manuscript.

**RC1:** 31: There is also an large collection of studies at the intermediate timescale, which isleft out here, dealing with surges. For sake of completeness, this might be included

**AC1:** We do reference a paper on surge dynamics by Yasuda and Furuya, but we have tried to keep the focus of this manuscript primarily on cyclic behavior.

**RC2:** This might be an ideal place to explain more in detail in what aspect you differentiate from others. In that way, referencing to other work is not needed that much.

**RC1:** 37: "the logistical challenge of" what is meant here?

**AC1:** We have modified the sentence to now read, ...due in part to the technical challenge of working with optical data in polar regions, where the surface is not touched by sunlight for months-long periods each winter. The technical challenge of working with optical data in polar regions is that several months go by each winter when optical data are not collected, because the sun does not illuminate the surface during those months. The full text of this manuscript provides a detailed description of the technical challenges of working with this data.

**RC1:** 39: "robust extraction" using a robust pre-processing technique, is different from a robust methodology. Given its stiffness (non adaptive) towards one model (a sinusoidal) this might not be a correct formulation. It might be "precise"...?

**AC1:** Following this suggestion, we have changed the word robust to precise.

**RC1:** 40: "primarily focused on Antarctica" maybe better: on the ice sheets and their outlets....?

**AC1:** We designed the study with the primary goal of understanding Antarctic seasonal ice dynamics, because thats where data is most limited, and its where the fewest studies have been published about the subject. The examples we provide and the statistics in all of the synthetic tests are based on Antarctic data. Accordingly, we stand by the statement that Our study is primarily focused on Antarctica, where seasonal variability is poorly understood, and where data limitations currently present the greatest challenges to making such measurements.

**RC1:** 53: "by feature tracking" add: over longer time spans

**AC1:** We have made the suggested change.
**RC1:** 55: "the true magnitude" change to "a" instead of "true" or "a well fit"

**AC1:** We have removed the phrase true magnitude, as suggested. The sentence now reads, ...by fitting a cyclic function to the time series of displacements rather than average velocities, we show that it is possible to accurately recover the magnitude and phase of seasonal velocity variability.

**RC2:** include "typical" or "general" as well, or some alike.

**RC1:** 56: I miss another possibility here, which is common practice in inSAR displacement estimation, being inversion (e.g.: Bontemps et al. 2018, 10.1016/j.rse.2018.02.023, Li et al. 2020, 10.1016/j.rse.2020.111695). This does not make it necessary to work with average time stamps or a model, and can resolves to very small time steps.

**AC1:** It is unclear how the inversion techniques described by Bontemps et al. or Li et al. should be included in this study.
**RC2:** Model-free time-step inversion seems worthwhile mentioning.

**RC1:** 64: " first or second image" first and second? maybe be more clear or use "master-slave" "chip-search space" etc.

**AC1:** We note that the community is moving away from the terms master and slave (`https://comet.nerc.ac.uk/about-comet/insar-terminology/`). The sentence in question states, Each satellite image may serve as the first or second image in multiple image pairs... The words "first" and "second" refer to the sequence in time. The suggested wording does not adequately convey temporal sequence, and the meaning of "chip-search space" may not be widely understood.

**RC1:** 100: "most robust means" why is this robust, where do you get the reliability from?

**AC1:** We have removed the word "robust".

**RC1:** fig4: maybe it is good to note, why there are two groups of points, as one is +- a year and the other at short time intervals in summer. btw: the purple line nicely follows the annual velocity clusters!

**AC1:** Following this suggestion, we have edited the caption of Figure 4 to describe the two groups of points as follows: The clustering of these 14,208 measurements taken near the grounding line of Byrd Glacier typifies ITS_LIVE image pair data, with short $\Delta t$ measurements providing direct, but noisy observations of velocity variability throughout each summer, while much lower-noise winter estimates can only give insight into the total displacement that occurs during the dark, winter months.

**RC1:** 151: I dont think this is sensitivity, but more an analysis to get an idea how good the recovery is. As the model is corrupted with noise and then an attempt is made to reconstruct the model. If I understand correctly.

**AC1:** The Sensitivity Analysis section is where we test the sensitivity of the method to several different parameters. We determine the sensitivity of the technique to the number of image pairs used, the level of background interannual variability, the amplitude of the underlying signal, and the phase of the underlying signal. The parameters of these tests are tabulated in Table 1: Sensitivity test parameters.

**RC1:** 181: "recover", change to we "are able to describe the variation by seasonal

cycles" or alike.

**AC1:** The passage in question reads, We conducted several tests to determine the accuracy with which we can recover the amplitudes and phases of seasonal cycles in synthetic velocity time series. We feel that the present wording will be more easily understood than the suggested wording.

**RC2:** You are not "recovering", you "describe" as you argument yourself. Please change.

**RC1:** 258: "is remarkable" subjective wording, please change

**AC1:** Following this suggestion, we have changed the word remarkable to notable.

**RC1:** 258: "robust", precise/accurate might fit better

**AC1:** Following this suggestion, we have changed the word robust to precise.

**RC1:** 267: "minuscule variations" subjective wording

**AC1:** We have replaced the word minuscule with tiny, to describe the displacements on the order of a meter or so that can be detected with this method.

**RC1:** 283: " in the climatological sense, nature does not consistently time such events as calving or increases in basal water pressure with any greater precision than the method we have presented to detect them". What is meant here?

**AC1:** We have added a definition of climatology to clarify that the climatology refers to the average seasonal cycle taken over many years. Transient events such as calving or impulses of water into a subglacial hydrological system often occur on a yearly cycle, but the corresponding glacier speedup may only last for a few days. It may seem that a spike in velocity only lasting a few days of the year would be poorly represented by a sinusoid that continuously varies throughout the year, but this sentence points out that mother nature does not time glacier calving to occur on the very same day each year. As a result, when we take the average annual cycle from many years of data, we find there is generally a season of glacier acceleration rather than just a few days of acceleration. To illustrate this point, here we consider a glacier whose velocity is exactly 800 m/yr most days of the year, but each summer it accelerates by 5010 m/yr for a duration of 103 days, centered on June

120 days. We generate 1000 years of this pattern, and then consider the mean cycle of daily speeds and compare it to the sinsuoid fit. Using the Matlab code provided at the bottom of this document, we get the following plot: What we see in the daily mean velocity curve is that in the climatological sense, the period of high summer velocity lasts for months, even though the average high-velocity event only lasts for 10 days. Fitting a sinusoid to the daily means finds a peak velocity on June 1, which is exactly the prescribed center date of the high-velocity period. Of course, the amplitude is severely underestimated by the sinusoid in this scenario, but the passage in question regards timing, not amplitude. It states, While it is true that a glacier can accelerate in response to a transient event and return to an equilibrium velocity within just a few days (Stevens et al., 2015; Andrews et al., 2014), in the climatological sense, nature does not consistently time such events as calving or increases in basal water pressure with any greater precision than the method we have presented to detect them.

**RC1:**Given your elaborate answer, you might agree with me, this sentence can be improved considerable . Now it is a very confusing sentence, so please put an effort on rewording.

---

## Author Response (AR3)

[revised manuscript text omitted]

**L9. "climatological" would be acceptable if you had already applied the method over > 30 yr of data. "Decadal" or "10-yr" average velocity is a more appropriate way to describe in the abstract the period over which the method has been tested and validated (I understand there is in principle not technical hurdles to apply the method to the full Landsat archive, spanning > 30 yr, but this is has not been done yet, at least not in this paper).**

We have changed the word *climatological* to *decadal*.

**L42. "global ice climatology" is ambiguous. I suggest "global ice dynamics" in this context.**

We have changed the word *climatology* to *dynamics*.

**Figure 1. Lines are thin and thus difficult to see. Make them thicker, especially the dashed black sinusoid.**

We have increased the thickness of all lines in Figure 1 by 50%.

**L62. "climatological" is OK here given the > 30 periods of the Landsat archive. So there is indeed a potential to derive a "climatology" using your method. No change required.**

Excellent.

**L68. "see" is not needed.**

Removed.

**L216. Add "(g-j)" here, just for clarity**

We have added *(g-j)*.

**L220. Add "(i,j)" here**

We have added *(i-j)*.

**L270. A reference would be welcome to back up the dramatic variability for PIG and Thwaites (Gourmelen, Mouginot or from others maybe?)**

We have added a citation to Mouginot et al., 2014.

**L290. "climatological" here suggests over 30-yr of data are needed before these short-term events are smoothed out. However, your case studies (Byrd, Russell) suggest that 10-yr of data is sufficient . What about "decadal average"?**

We have changed *climatological* to *decadal*.

**L303. This statement suggest "several decades" (at least this is how I received it). Here I wondered whether there are enough Landsat image pairs between 1985 and 2000 so that the method can be applied and would also work? I think it is reasonable for a reader to ask this, because your Byrd and Russell glacier examples start both around 2007 / 2008. If the frequency of image pairs is as high throughout the Landsat archive then the statement can be left unchanged. Otherwise, authors should adjust the wording.**

Ah, yes, we see your point. Our previous wording could be interpreted to mean that a sufficient quantity of data are available to give insights into seasonal dynamics during the period of 1985-2000. At Russell Glacier and most of coastal Greenland the usable record begins in 2000 (we started our Russell analysis in 2007 for the most direct comparison to the GPS record). We've removed the suggestion that insights into ice climatology can be gained as far back as 1985. The section now reads:

*...our method might not only meet, but exceed the performance of the in situ GPS receiver while providing insights into **dynamic behaviors that occurred years before the station was installed***.

---

## Author Response (AR4)

**Dear Authors,**

**Thanks for taking into account my minor comments on the text.**

**I am delighted to accept your paper for publication in our journal. Thank you for choosing TC to publish your work.**

**My only remaining comment concerns Figure 1. The two sinusoids are very thin and hard to see (at least for me!). This is not an error so you [can] keep it "as is" (I will not check) but the figure would gain readability by using thicker lines, and it is not an important figure after all.**

**Best regards,**

**Etienne Berthier**

Thanks for giving the manuscript another close look. For line widths in Figure 1 and all other figures in this paper we're following the guidelines provided by the Nature journal group (https://www.nature.com/gim/authors-and-referees/figures-tables-and-artwork-guidelines).

By setting line transparency we've intentionally made sure the sinusoids in Figure 1 do not dominate the figure or compete visually with the primary information (the blue true velocity time series) or the secondary information (the four image acquisition times and corresponding velocities estimates obtained by the six unique image pairs).

The sinusoids are intentionally faint as their only purpose is to help the reader imagine what it might mean to fit a sinusoid to the observed velocities, and how that would compare to a sinusoid fit to the true signal. Our first attempts at creating this figure did not exploit transparency, and the overall effect was a jumbled mess of lines that buried the main intention of the figure, which is to explain how velocities are obtained from displacements measured in image pairs.

In the interest of keeping the main message of the figure at the center of focus, we've chosen to keep the figure as is.